# Understanding the Compound Flood Risk along the Coast of the Contiguous United States

Dongyu Feng[1], Zeli Tan[1], Donghui Xu[1], L. Ruby Leung[1]

[1]Atmospheric Sciences and Global Change Division, Pacific Northwest National Laboratory, Richland, WA, 99354, USA

*Correspondence to*: Zeli Tan (zeli.tan@pnnl.gov)

**Abstract.** Compound flooding is a type of flood events caused by multiple flood drivers. The associated risk has usually been assessed using statistics-based analyses or hydrodynamics-based numerical models. This study proposes a compound flood (CF) risk assessment (CFRA) framework for coastal regions in the contiguous United States (CONUS). In this framework, a large-scale river model is coupled with a global ocean reanalysis dataset to (a) evaluate the CF exposure
related to the coastal backwater effects on river basins, and (b) generate spatially distributed data for analyzing the CF hazard using a bivariate statistical model of river discharge and storm surge. The two kinds of risk are also combined to achieve a holistic understanding of the continental-scale CF risk. The estimated CF risk shows remarkable inter- and intra-basin variabilities along the CONUS coast with more variabilities in the CF hazard over the US West and Gulf coastal basins. Different risk assessment methods present significantly different patterns in a few key regions, such as San Francisco Bay
area, lower Mississippi River and Puget Sound. Our results highlight the needs to weigh different CF risk measures and avoid using single statistics-based or hydrodynamics-based CFRAs. Uncertainty sources in these CFRAs include the use of gauge observations, which cannot account for the flow physics or resolve the spatial variability of risks, and underestimations of the flood extremes and the dependence of CF drivers in large-scale models, highlighting the importance of understanding the CF risks for developing a more robust CFRA.

Keywords: compound flood, coastal flood, flood risk assessment, uncertainty analysis, river model

## 1 Introduction

Compound flooding is a type of multivariate flood events when various flood drivers occur concurrently in the same or adjacent regions (Santiago-Collazo et al., 2019). Specifically, over coastal regions, compound flooding is generally driven by
fluvial and coastal processes. While an individual driver may not be extreme, the complex nonlinear interactions between fluvial and coastal processes can intensify the joint impact of multivariate drivers (Dykstra & Dzwonkowski, 2020), causing significant flood hazards (Mehran et al., 2017; AghaKouchak et al., 2018) and negative socio-environmental impacts (Hinkel et al., 2014; Wahl et al., 2017). It is possible that a compound flood (CF) event is not caused by extreme weather (Couasnon et al., 2020) but rather occurs when one or multiple flood drivers exceed their respective thresholds (Zscheischler et al.,

2020). Assessing CF caused by co-occurring fluvial and coastal flooding is important for low-lying coastal regions where 680 million people live globally and this number is projected to increase to over 1 billion by 2050 (Pörtner et al., 2019). Such flood hazard is intensified during "wet" storms by simultaneous rainfall and storm surge events and can be exacerbated by future sea level rise (Kulp and Strauss, 2019) and climate change (Bevacqua et al., 2019; Gallien et al., 2018; Gori & Lin, 2022). To mitigate the CF risks, it is crucial to understand the driving processes and the related uncertainties in the risk

assessment.

Compound flood risk assessment (CFRA) is critical for flood planning, management, timely emergency response and decisions. CF risk has substantial spatial variabilities since the CF drivers and the CF risk dependence on the drivers are affected by the local conditions (Wahl et al., 2015), such as the characteristics of local basins that affect runoff generation, river routing (Hendry et al., 2019), synoptic weather systems, and storm characteristics (Seneviratne et al., 2012). CFRA can

be classified into statistics-based and hydrodynamics-based approaches. Statistics-based CFRAs rely on statistical modeling and define the CF hazard as the frequency of a CF event. Hydrodynamics-based CFRAs use numerical simulations that can represent human exposure to CF events with considering the spatiotemporal variabilities and interaction of CF drivers.

At regional and global scales, statistics-based CFRAs consider the CF hazard as statistical dependence or co-occurrence rate of multiple flood drivers including discharge and surge (Moftakhari et al., 2017; Sadegh et al., 2018; Muñoz et al., 2020),

precipitation and surge (Bevacqua et al., 2019), discharge, surge, and wave (Camus et al., 2021), etc. Statistics-based CFRAs perform statistical analysis using long-term data at paired gauges near the land-ocean interface. The data can be obtained either from large-scale numerical simulations (Eilander et al., 2020; Nasr et al., 2021) or gauge observations (Ward et al., 2018; Paprotny et al., 2020). Bivariate or multivariate analyses are performed to measure the CF hazard in terms of the joint occurrence of event extremes (Salvadori et al., 2007; Zscheischler et al., 2020). The CF hazard is determined either using the

extreme dependence among multiple CF drivers or the likelihood of their joint occurrence. The dependence structure can be assessed from correlation and/or tail dependence coefficients (Wahl et al., 2015; Nasr et al., 2021). The co-occurrence rate may be calculated as the joint exceedance probability when a single or multiple drivers are above their predefined thresholds (Moftakhari et al., 2017), e.g., 95th or 99th percentile (Kew et al., 2013), which are defined respectively as "OR" and "AND" hazard scenarios by Salvadori et al., 2016.

Statistics-based CFRAs can reveal critical regional variability in terms of the strength of individual drivers, their dependence structures, and joint occurrence, as well as the CF hotspots. Gauged observations provide a robust basis for large-scale risk assessments (Couasnon et al., 2020). The simple structure in statistical models facilitates the investigation of major CF drivers. However, the variability of CF risks is limited to the gauge level since data of the entire river basin is often unavailable. Consequently, the physical processes behind flood drivers and the influence of local basin characteristics and

river topology cannot be fully explored.

CF risk can vary substantially across different rivers and estuaries (Xiao et al., 2021; Zhang et al., 2020) as a result of the impact of river topology and tidal variations (Bakhtyar et al., 2020; Gori et al., 2020) and the characteristics of drainage basins that regulate the river processes (Dykstra & Dzwonkowski, 2021). For example, river topology controls streamflow

routing and backwater propagation through river networks (Bilskie & Hagen, 2018). Even if the statistics-based CFRA

yields a high probability of CF event in a region, the CF exposure can be limited by a steep channel slope because the coastal backwater is not able to propagate upstream. Thus, the physical processes can influence the results of CFRA. Hydrodynamics-based CFRAs have been applied to measure the population and property exposure to CF, i.e., the CF exposure, using spatially abundant observations (Dykstra & Dzwonkowski, 2020; Valle-Levinson et al., 2020), numerical models (Kumbier et al., 2018; Lian et al., 2013; Olbert et al., 2017; Ye et al., 2020), and the integration of both (Moftakhari

et al., 2019; Muñoz et al., 2020; Serafin et al., 2019). However, applications of hydrodynamics-based CFRAs are mostly limited to basin scales because of the computational cost of high-resolution numerical models.

Recent developments in large-scale river models (Feng et al., 2022; Ikeuchi et al., 2017; Luo et al., 2017) and global water level and storm surge reanalysis datasets (Muis et al., 2017, Muis et al., 2020) facilitate hydrodynamics-based CFRA across rivers and estuaries. Large-scale river models can capture streamflow at fine temporal scales (Towner et al., 2019) and

resolve backwater effects when coupled with the tide and surge induced water level (Feng et al., 2022; Muis et al., 2020). Such models offer the capability to evaluate spatiotemporally varied CF drivers, flood extent and population exposure to CF events from basin to global scales over multiple decades (Ikeuchi et al., 2017; Eilander et al., 2020; Eilander et al., 2023).

The CF hazard and exposure evaluated separately by the aforementioned statistics-based or hydrodynamics-based CFRAs may produce inconsistent results (K. Xu et al., 2022). The risk determined based on either CFRA can cause biased

judgments. For example, for high-gradient and sparsely populated regions, high CF hazard will not result in high CF exposure. Instead of advocating for either method, this study proposes a CFRA framework that analyzes both hazard and exposure, as well as the CF risk that combines the two types of risks (Kron, 2005). We identify the strengths and limitations of each framework and highlight the possible uncertainties within the CFRA framework.

A robust CFRA should consider the uncertainties associated with frequency and possible damages of compound flooding

and provide a thorough understanding of the uncertainties related to the risk analysis (Apel et al., 2004). The uncertainty can stem from various sources in both statistics-based and hydrodynamics-based CFRAs, such as measurement errors and approximations in numerical models. A comprehensive understanding of the uncertainty sources in CFRA is crucial for managing and predicting CF risks and will provide valuable insights for guiding future improvements. Uncertainty analysis is challenging due to various uncertainty sources and has drawn significant attention in risk assessments in the fields of

coastal flooding (Hinkel et al., 2014; Vousdoukas et al., 2018; Parodi et al., 2020), fluvial flooding (Apel et al., 2004; Egorova et al., 2008), and compound flooding (Dung et al., 2015; Sadegh et al., 2017; Sadegh et al., 2018; Zhang et al., 2020).

The contiguous United States (CONUS) (Fig. 1) consists of 48 states, with coastal counties occupying about 10% of the total area. There are 17 major port cities, and ~40% of the US population residing in coastal counties are subject to high coastal

flooding risks (Hanson et al., 2011). A high-resolution analysis study including pluvial, fluvial, and coastal flooding, projected a significant changing pattern of the flood risk in CONUS under future climate scenarios (Bates et al., 2021). Particularly, the CF risk was previously evaluated for the CONUS coastline or major US coastal cities using statistics-based

CFRAs in terms of the dependence between storm surge and precipitation (Wahl et al., 2015), seasonable dependence among multiple CF drivers (Nasr et al., 2021), and the joint probability in "OR" hazard scenarios in response to sea level rise

(Moftakhari et al., 2017). However, most existing studies rely on the CF driver measured/modeled at a single site and have not accounted for the dynamic change of river flow, such as the spatiotemporally varied streamflow, as well as river topology, coastal backwater effects and the associated uncertainties. The CF exposure is also poorly understood.

The objectives of this study are threefold: (a) to develop a new CFRA framework based on both statistical analyses and a large-scale river model that is coupled with a global ocean model reanalysis product; (b) to provide a holistic hazard and

exposure risk assessment of the compounding fluvial and coastal flooding along the CONUS coastline, and (c) to understand the uncertainties in both statistics-based and hydrodynamics-based CFRAs.

## 2 Methodology

This section describes the new CFRA framework and provides details of the statistical and river modeling approaches. We also describe the methods to identify uncertainties within the CFRA.

**2.1 The CFRA framework**

The CFRA framework (Fig. 1) provides estimates of CF hazard, exposure, and the overall risk (Maskrey et al., 2011). The CF hazard refers to the temporal frequency of CF events and is derived from the bivariate statistical modeling of river discharge and storm surge. The CF exposure is defined as the exposed population within the CF backwater extent, which is modeled using a large-scale river model, the Model for Scale Adaptive River Transport (MOSART) (Li et al., 2013).

Correspondingly, the CF risk is the combination of the hazard and exposure.

MOSART is a physics-based river routing model which can be applied at the basin to global scales. The model takes the total runoff generated by a land surface model and routes the surface runoff from hillslope to tributary subnetworks, which along with the subsurface runoff are discharged to river outlets through the main channels. In this study, kinematic wave method is used for overland flow routing, whereas diffusive wave method is applied in the river channels to represent the

coastal backwater effects (Feng et al., 2022). The MOSART simulation is performed on the 1/8° resolution CONUS grid. The MOSART configuration on the same grid has been validated and applied to flow and sediment simulations (Li et al., 2015a; Li et al., 2022). The model parameters are available globally with more detailed descriptions in previous studies (Li et al., 2013; Li et al., 2015b). The model is run at an hourly time step from 1979 to 2018 and daily outputs are archived for analysis. The first-year simulation is excluded for analysis due to model spin-up. The floodplain inundation is represented

using a macroscale inundation scheme (Luo et al., 2017). The channel slope and the riverbed elevation are derived from the 15 arcsec digital elevation model (DEM) of the HydroSHEDS and river vector data (Lehner et al., 2008, Lehner & Grill, 2013). The runoff forcing is from Global Reach-level Flood Reanalysis (GRFR) (Yang et al., 2021), a bias-corrected offline simulation from a high-resolution VIC land surface model forced with precipitation from the Multi-Source Weighted-

Ensemble Precipitation (MSWEP) (Beck et al., 2019) and other climatic forcings from ECMWF Reanalysis v5 (ERA5) (Hersbach et al., 2018). The GRFR forcing has shown excellent performance of simulating extreme streamflow events (Yang et al., 2021). The downstream boundary is enforced at the river outlets for rivers with a drainage area >1000 km$^2$ (169 in total). We apply two types of boundary condition (BC): (1) time-varying storm surge (*SS*) level and (2) fixed mean sea level. Both are obtained from the third-generation Global Tide and Surge Model (GTSM) (Muis et al., 2022). The *SS*-induced backwater effects in this study are quantified by comparing the two simulations which use the first and second BCs, respectively (Feng et al., 2022). For small river basins, we apply the normal depth boundary at their outlets (Feng et al., 2022), which is MOSART's default setting. The GTSM is a global hydrodynamic model with a coastal resolution of ~2.5 km (~1.25 km in Europe). Driven by the ERA5 atmospheric reanalysis dataset, the GTSM produces time series of hourly total water level and storm surge at global coasts from 1979 to 2018 (Muis et al., 2020). The total water level in GTSM has been validated globally against gauged measurements (Muis et al., 2020) and the storm surge has been validated using historical events driven by tropical cyclones and extratropical cyclones (Dullaart et al., 2020).

The modeled streamflow used in the statistical analysis is from MOSART simulation forced by the dynamic GTSM BC. The MOSART simulated streamflow is validated at 61 USGS gauges (Fig. 2), which are selected based on the following criteria: (1) These gauges are located at the mainstem of the rivers and within 80 km of the corresponding river outlets; (2) the corresponding river reaches have upstream drainage areas larger than 1,000 km$^2$; and (3) the gauge data have a temporal coverage longer than 10 years. The MOSART accuracy is evaluated using Kling-Gupta efficiency (KGE) (Gupta et al., 2009) and coefficient of determination ($r^2$). MOSART shows reasonable accuracy in simulating daily streamflow (Fig. S1), with both $r^2$ and KGE generally over 0.6. The model performance is lower at a few gauges, likely caused by the coarse grid resolution (1/8°), approximations of river geometry in MOSART, and uncertainty in the GRFR runoff data. The GTSM simulated water level along the CONUS coastline is validated against the NOAA measurements at 34 tidal gauges (Rashid et al., 2019) that have 80% or more data available over the simulation period (Fig. 2). The GTSM modeled water level achieves satisfactory performance along the CONUS coast when measured by $r^2$ and root mean squared error (RMSE) (Fig. S2): $r^2$ is generally over 0.75 and RMSE is below 0.5 m. There are only two exceptions on the East coast with either low $r^2$ or high RMSE, because the GTSM grid of the two gauges does not resolve the corresponding estuaries. In the context of constructing a new CFRA framework within the CONUS domain and investigating the associated uncertainties, the performance of MOSART and GTSM models is deemed satisfactory in large-scale simulations.

The CF hazard is derived from the bivariate statistical modeling. The analysis is performed for the MOSART coastal cells which are defined as the grid cells within seven upstream cells from the corresponding river mouths. It is assumed that coastal processes have no impacts on the regions beyond this extent. The simulated daily streamflow ($Q$) at each selected cell is paired with the daily maximum storm surge (*SS*) level from the GTSM reanalysis dataset at the grid cell nearest to the outlet. The CF hazard is calculated by the following procedure:

(a) CF event selection: use a $SS$ event selection scheme (Feng et al., 2022) to extract all $SS$ events with the $SS$ level over 95th percentile and then in the selected SS events identify them as CF events if river discharge of the corresponding station during these events is also over 95th percentile;

(b) univariate analysis: fit the selected $SS$ and $Q$ into their marginal distributions and calculate the marginal exceedances, i.e., the probability of exceeding the 95th percentile of the respective marginal distributions;

(c) dependence assessment: determine if the bivariate variables are dependent of each other based on Kendall's rank correlation coefficient ($\tau$) (Kendall, 1938);

(d) bivariate analysis: calculate the joint exceedance probability based on "AND" hazard scenario (Salvadori et al., 2016) that accounts for both marginal distributions and dependence structure.

As the first step, our event selection scheme samples independent $SS$ events from the time series data, which avoids dependence in the extremes and eliminates the need for declustering. We assume that the $Q$ extreme within each $SS$ event is independent as both frequency and duration of $SS$ are generally much smaller than that of fluvial flooding. While it is possible that the duration of a fluvial flood event does not precisely align with a $SS$ event, we don't include any time lag between $Q$ and $SS$ in the consideration as this study specifically quantifies the CF impact based on the $SS$-driven backwater effects. The threshold of 0.95 ensures that at least 50 pairs of $Q$ and $SS$ data points are available for bivariate modeling. The occurrence probability of the storm surge events ($P(SS)$) is calculated for each river basin as the ratio of the duration of all $SS$ events divided by the simulation period.

In the univariate analysis, the marginal distributions of $Q$ and $SS$ ($f_Q$ and $f_{SS}$) are selected based on the AIC (Akaike Information Criterion) statistics from 8 candidate distributions: Gamma, Generalized Pareto, Pearson Type III, Lognormal, Generalized Extreme Value, Generalized Logistic, Log-gamma and Gumbel. The fitted distributions are tested using the Kolmogorov-Smirnov and chi-square tests for goodness of fit. The marginal exceedance probabilities of $Q$ and $SS$ ($P_Q$ and $P_{SS}$) are

$$P_Q = F_Q(q^*), \tag{1}$$

$$P_{SS} = F_{SS}(ss^*), \tag{2}$$

where $q^*$ and $ss^*$ represent the 95th percentile values.

The dependence between $Q$ and $SS$ is assessed for each MOSART cell by calculating the Kendall's correlation ($\tau$). The significance level is set as 0.05 following previous studies of statistics-based CFRAs (Ghanbari et al., 2021; Moftakhari et al., 2017). We consider $Q$ and $SS$ to be dependent of each other when they display a significant positive correlation (p-value<0.05). Although assessed in extensive CF literature, the dependence structure alone does not represent the CF hazard. For example, in a case when $Q$ and $SS$ are highly dependent, the CF risk can still be low if both drivers do not show frequent extremes. Thus, the joint exceedance probability is calculated based on the "AND" hazard scenarios (Salvadori et al., 2016), which assumes both $Q$ and $SS$ exceed their corresponding thresholds.

The joint exceedance probability ($P_{Q,SS}$) is given as

$$P_{Q,SS} = 1 - F_Q(q^*) - F_{SS}(ss^*) + F_{Q,SS}(q^*, ss^*), \tag{3}$$

where $F_{Q,SS}$ is the cumulative joint distribution which is a function of the cumulative marginal distributions, $F_Q$ and $F_{SS}$. When $Q$ and $SS$ are independent, $F_{Q,SS}(q^*, ss^*)$ is simply the product of the marginal exceedance probability:

$$F_{Q,SS}(q^*, ss^*) = F_Q(q^*) \cdot F_{SS}(ss^*). \tag{4}$$

When $Q$ and $SS$ are dependent, the joint distribution can be expressed using a copula function of the marginal distributions as (Grimaldi & Serinaldi, 2006):

$$F_{Q,SS}(q^*, ss^*) = C_{Q,SS}(F_Q(q^*) \cdot F_{SS}(ss^*)), \tag{5}$$

where $C_{Q,SS}$ is the bivariate copula function that allows the analytical formulation of the dependence structure. For each MOSART coastal cell where $Q$ and $SS$ are dependent, the copula function is selected from 24 candidate families following the instructions provided in Moftakhari et al., 2017, using the R-package "copula" (Kojadinovic & Yan, 2010). The marginal exceedance probabilities ($P_Q$ and $P_{SS}$) and their joint exceedance probability ($P_{Q,SS}$) are conditioned on the occurrence of the storm surge events as they are calculated from the SS data sampled in Step (a). These probabilities are multiplied by $P(SS)$ to obtain the unconditional probabilities.

The CF exposure is defined as the accumulated population ($W_p$) over the coastal backwater flooded region during CF events when $Q > q^*$ and $SS > ss^*$. To calculate $W_p$, we use the 1000-m resolution Global Human Settlement Layer (GHSL) population data that is updated every five years from 1975 to 2020 (Schiavina et al., 2019). We aggregate the data to the 1/8° MOSART grid and linearly interpolate the data over the simulation period. The backwater flooded fraction caused by CF is identified by comparing the simulations with the two different downstream BCs:

$$\Delta f(t, i) = f_{GTSM}(t, i) - f_{MSL}(t, i), \tag{6}$$

where $f$ represents the simulated flooded fraction of each grid cell, $t$ is the model output time step during CF and $i$ is the grid cell index of the MOSART coastal cells. During a single CF event, human exposure in a grid cell is the product of the corresponding population and $\Delta f$. Thus, the CF exposure is the accumulated human exposure over all CF events during the simulation period.

The CF risk is represented by a risk index ($CFRI$), defined as the product of the CF exposure and the CF hazard (Judi et al., 2018; Kalyanapu et al., 2015; Phongsapan et al., 2019):

$$CFRI = CFHI \times CFEI, \tag{7}$$

where the CF hazard index ($CFHI$) and the CF exposure index ($CFEI$) are calculated by normalizing $P_{Q,SS}$ and $W_p$ with their corresponding 95[th] percentile values. We do not use the maximum value as the normalizing constant because the maximums can be too extreme and likely concentrates on the river outlets where $CFEI$ is high. The use of such a normalization would shadow the $CFRI$s at upstream regions. Our approach integrates measures of risks that consider both the probability of occurrence and human exposure. However, it should be noted that the combination of different types of risks, despite providing a comprehensive estimation of the CF risk, is subjective and may affect the risk assessment results.

## 2.2 Uncertainty analysis in CFRA

In this section, we review the uncertainty sources in the large-scale CFRAs. The uncertainty analysis is critical for a robust CFRA. Given that uncertainty can arise from diverse sources in both statistical and numerical models, an improved understanding of the uncertainty sources will provide valuable guidance for the future enhancement of the CFRA framework. We first examine the spatial variability in streamflow and storm surge and the relative impacts of riverbed elevations on the hydrodynamics-based CFRA, as neglecting these physical factors leads to uncertainties in the statistics-based CFRA. The uncertainties in the statistics-based and hydrodynamics-based CFRAs are also assessed by comparing the risk estimates at paired observation gauges. While it is challenging to accurately quantify such uncertainties in the CFRAs, we aim to highlight the significance of different uncertainty sources.

### 2.2.1 CFRA uncertainty sources

The uncertainty in CFRA can generally be classified into two categories: aleatory uncertainty and epistemic uncertainty. Aleatory uncertainty is inherent to the intrinsic variability in natural and anthropogenic systems (Hall, 2003). Epistemic uncertainty is due to limited knowledge of natural systems and can be reduced with an improved understanding of the systems (Ferson & Ginzburg, 1996; Uusitalo et al., 2015). Herein we list and classify the possible uncertainty sources in the CFRA (Table 1). This classification may be subjective because sometimes the distinction between incomplete knowledge of the systems and natural variabilities cannot be easily identified (Apel et al., 2004).

In statistics-based CFRAs, aleatory uncertainties are related to the spatial variabilities of the fluvial processes and river topology that are not well represented in gauge data (Fan et al., 2021) and the stationary assumption of statistical models (Ghanbari et al., 2021). It is widely known that the CF risk is nonstationary due to the changing climate. Additionally, the flood drivers vary significantly depending on the local topology (Sun et al., 2021), which is usually not accounted for in statistics-based CFRAs. The timing of peak floods changes from an upstream gauge to the outlet and the storm surge varies between an offshore tidal gauge and the river mouth. Although the statistics-based CFRAs use a time window of 1~5 day (Ward et al., 2018; Wu et al., 2021) to account for this time lag, this procedure inevitably increases the possibility of falsely matching two independent events when distant observation gauges are used. Epistemic uncertainties in statistical models can include measurement errors and model structure uncertainty. Although the errors of the water level measurements at National Oceanic and Atmospheric Administration (NOAA) tidal gauges are usually small (O(1mm)) (Asher et al., 2019), the quality of the U.S. Geological Survey (USGS) measured streamflow varies significantly. For example, it was found that the USGS streamflow errors can reach over 8% (Turnipseed & Sauer, 2010), and be even much larger during extreme events as the measurements are not sufficiently continuous to cover many extremes (Kiang et al., 2018). Moreover, USGS gauges may not be installed exactly at the river-ocean interface, which cannot capture the river discharge to the ocean. The statistical analysis of CF risks based on these measurements will inevitably be biased. Moreover, model structure uncertainties always

exist in statistical models, such as the selection of marginal distribution functions and dependence on level of significance. The latter is critical for computing the joint exceedance probability of CF (Fan et al., 2021).

Numerical models used in both statistics-based and hydrodynamics-based CFRAs also have many uncertainties. Intrinsically, there is uncertain climate change that modifies climatological and societal systems (Bouwer, 2013). For numerical models, the epistemic uncertainty can be classified into uncertainties in model structure, model parameters and data. The parameter uncertainty and the data uncertainty are caused by uncertain model parameters (e.g., channel roughness coefficient), uncertain river topology, and channel geometry, respectively. In large-scale river models, the hydraulic physics are usually simplified to guarantee computational efficiency, such as using an empirical formulation of floodplain inundation (Yamazaki et al., 2012), approximations in flood wave physics (Hodges, 2013). In addition, coarse mesh resolutions used by large-scale river models can cause unresolved river networks and topology (Parodi et al., 2020). All these uncertainties could be related to inaccurate assessments of the event extremes (Muis et al., 2017) and flood drivers' dependence (Nasr et al., 2021).

### 2.2.2 Impact of riverbed elevation

The riverbed elevation determines the extent of coastal backwater propagation. To understand its impacts on CF risks, a random forest analysis (Breiman, 2001) is performed to evaluate the relative importance of riverbed elevation against $Q$ and $SS$ to the backwater effects.

Random forest models are widely used to assess the relative importance of predictors with respect to a response variable (Breiman, 2001; Woolway et al., 2021). Here, the predictor variables are $Q$, $SS$ and the riverbed elevation. For each coastal grid cell, we use the MOSART simulated $Q$, the GTSM simulated $SS$ at the river outlet, and the grid cell elevation. The response variable is the backwater-induced water volume change ($\Delta V$):

$$\Delta V(t,i) = \big(h_{GTSM}(t,i) - h_{MSL}(t,i)\big)L(i)W(i) + (fv_{GTSM}(t,i) - fv_{MSL}(t,i)), \tag{8}$$

where $h$ is the channel water depth at the $i$-th grid cell and the time t, $L$ is the main channel length of the $i$-th cell and $W$ is the corresponding width, and $fv$ is the floodplain water volume. The predictor and response variables are normalized to [0,1] before fitting into the random forest model. We fit independent random forest models for every coastal river basin, with sizes varying from ~10 grid cells to >100 cells.

### 2.2.3 Impact of fluvial processes

The impact of complex fluvial processes on streamflow is significant. The lag in streamflow peaks between an upstream location and the river outlet will cause biases in the CFRA if the upstream $Q$ measurements are used in the CF risk analysis. To identify the associated uncertainty, we compute the time-averaged shift of modeled $Q$ or $SS$ peaks between the observation gauges and the corresponding river outlets over the simulation period. The calculation of the shift in peaks includes the following steps: (i) for a USGS or NOAA gauge, we first locate the MOSART or GTSM grid cell nearest to the

corresponding river outlet and extract the $Q$ or $SS$ extreme events of the grid cell; (ii) we then identify the peak date of each extreme event and define a time window of 10 days around the extreme; (iii) over the defined time window, we search for the date of the peak simulated $Q$ or $SS$ at the MOSART or GTSM grid cell where the USGS or NOAA gauge is located and calculate the difference between the two peak dates. If no peaks are identified for the gauge grid cells, we assume that the difference is five days.

We also compute the flow time over the river basins, defined as the time that terrestrial runoff takes to travel from an upstream cell to the outlet via the river network. The flow time is determined by the basin characteristics, such as channel geometry, meandering and riverbed elevation. A longer flow time typically implies a larger time lag in streamflow peaks. In this study, the calculated flow time of each grid cell is averaged over the simulation period.

### 2.2.4 Model-data comparison

Lastly, we also evaluate the uncertainty caused by using measured versus modeled $Q$ and $SS$ for analysis. For this, we compare the statistical metrics of $P_Q$, $P_{SS}$, $\tau$ and $P_{Q,SS}$ for the modeled and measured pairs of $Q$ and $SS$ at 24 river basins (Table S1), where a USGS gauge paired with a neighboring NOAA tidal gauge can be found. We include three combinations of $Q$ and $SS$, as listed in Table 2. The comparison between the combinations (a) and (c) represents the uncertainty due to fluvial processes and river topology. The comparison between the combinations (b) and (c) represents the uncertainty from the numerical modeling.

## 3 Results

### 3.1 Uncertainty in CFRA

### 3.1.1 The relative importance of riverbed elevation

Our result shows the crucial role of the riverbed elevation (Fig. 3) in determining the CF risks in the river basins, which contrasts with previous studies that mostly focused on the dynamics of $Q$ and $SS$. In particular, the elevation effect dominates in the Northwest coast (Fig. 3a), where the large riverbed elevation impedes the propagation of coastal backwaters and the areas of high CF risks are thus restricted to the coastline. In the other regions, the relative importance of $Q$ and $SS$ varies, which also depends on the riverbed elevation (Fig 3b and 3c). The $SS$ impact is limited along the West coast due to the elevated river channels but exceeds the impact of $Q$ in the low-lying East and Gulf coasts. The importance of the riverbed elevation to CF risks identified in this study is consistent with findings from some local studies (Bilskie & Hagen, 2018, Gori et al., 2020). In brief, the relative importance of the CF drivers varies depending on local basin characteristics.

### 3.1.2 Shift in peaks

The shifted peak days in $Q$ and $SS$ from the USGS or NOAA gauges to the corresponding river outlet show that the statistics-based CFRA may have large uncertainties (Fig. 5 and 6). The averaged time lag in the $Q$ peaks varies from 1 to 5 days (Fig. 6), depending on the local topology, basin characteristics, and hydrological response to coastal backwaters. In the northwestern and northeastern river basins, where the elevation gradient is large, the flow time is $\sim$ 1 day and the resulting shifts are small ($\sim 1\pm1$ day). In contrast, the shifts and the flow time are much larger ($\sim 3\pm1$ days) in the low-gradient regions of the East and Gulf coasts. The time shifts in the $SS$ peaks generally depend on the distance between the tidal gauges and river mouth. In CONUS, the shifts are generally small ($\sim$ 1 day) with lower variabilities. Our results show that the combined shifts in the peaks of the two flood drivers in some locations can be greater than five days, a duration used by many previous studies as the time window to identify extreme CF events (Ward et al., 2018; Wu et al., 2021).

### 3.1.3 Model-data comparison

We compare the marginal exceedance probabilities of discharge and storm surge ($P_Q$ and $P_{SS}$), the Kendall's rank correlation coefficient ($\tau$) and the joint exceedance probability ($P_{Q,SS}$) computed from the three combinations of $Q$ and $SS$ described in Section 2.2.4 at 24 river basins (Fig. 7). For the same river basin, these statistical metrics can differ significantly among the combinations, indicating substantial uncertainties due to fluvial processes and river topology, as well as from the numerical modeling. Generally, $P_{SS}$ is more consistent among the three combinations, because the time shifts in the $SS$ peaks are small (Fig. 6). In contrast, $P_Q$, $\tau$ and $P_{Q,SS}$ show greater variations.

There are significant differences of $P_Q$, $\tau$ and $P_{Q,SS}$ between the combinations of the modeled $Q$ and $SS$ at the interface and at the observation gauges, particularly in the West coast (black vs. blue bars in Fig. 7). As discussed in Section 2.2.3, this indicates the spatial variabilities of the CF risk within the river basins and the associated uncertainty in the statistics-based CFRA. The uncertainty in $P_{Q,SS}$ varies along the CONUS coast and is more distinct in several basins (e.g., 14243000 and 11530500) due to the larger variability in $P_Q$ and higher $\tau$.

There are also significant differences of $P_Q$, $\tau$ and $P_{Q,SS}$ between the combinations of modeled and measured $Q$ and $SS$ at observation gauges across all CONUS coasts (blue vs. red bars in Fig. 7). As mentioned in Section 2.2.3, the differences indicate the uncertainty within the numerical models that could influence the assessment of the CF risk. The values of $P_Q$ and $P_{SS}$ calculated from the modeled $Q$ and $SS$ are generally smaller than those calculated from observations. This is likely because the MOSART and GTSM models underestimate the $Q$ and $SS$ extremes, a well-known uncertainty in large-scale models (Muis et al., 2017; Yang et al., 2021). It should be noted that the USGS reported streamflow peaks are likely uncertain because USGS derives streamflow based on the stage-discharge relationship, but the data used for the derivation are rarely collected during extreme events (Turnipseed & Sauer, 2010). Besides the uncertainties mentioned above, we find that the use of modeled $Q$ and $SS$ could lead to underestimation of the dependence ($\tau$) and thus the joint risk ($P_{Q,SS}$),

particularly in the West and Northeast coasts (Fig. 7). The uncertainties in $P_Q$ and $\tau$ are more important in determining the uncertainty in $P_{Q,SS}$ in the West and East coasts, respectively. The model-data comparison is provided for a few example basins to demonstrate the various types of uncertainties (see Supplement for further details).

The uncertainty analyses underscore the uncertainties in both statistics-based and hydrodynamics-based CFRAs, which
should be made aware of in applications. Importantly, while the uncertainty in the hydrodynamics-based CFRA may be reduced by improving the numerical models, it is not possible for the statistics-based CFRA to account for the physical processes and the variability at the basin scale, such as the varied streamflow and backwater propagation extent.

## 3.2 CFRA

This section shows the CFRA framework that provides spatially distributed CF risk estimates based on the modeled $Q$ and
$SS$ and captures the impacts of fluvial processes and riverbed elevation. The CF risk combines the CF hazard derived from statistical models and the CF exposure simulated by the coupled MOSART and GTSM.

### 3.2.1 CF hazard

The CF hazard is represented by the joint exceedance probability of $Q$ and $SS$ ($P_{Q,SS}$), which depends on their respective marginal exceedance probability ($P_Q$ and $P_{SS}$) and the dependence structure (Fig. 8 and 9). The spatial map shows larger $P_Q$
variations in the West and Gulf coasts but a more uniform $P_Q$ pattern in the East coast. The highest $P_Q$ ($\sim 3.0\%$) is observed in the Northwest coast (Fig. 8), where the corresponding $P_{SS}$ is low ($\sim 1.0\%$) (Fig. 9). The variability in $P_{SS}$ is much smaller compared to that of $P_Q$. The values of $P_{SS}$ are high ($\sim 2.0\%$) in the western Gulf coast in correspondence with the moderate $P_Q$ of the same basins ($\sim 2.5\%$). Moreover, $P_Q$ shows critical intra-basin variability within several basins, with a standard deviation of up to 1%. The marginal probability provides the basis to derive the drivers' dependence structure, copula
functions, and joint probability.

The result of the Kendall's rank correlation coefficient ($\tau$) shows large inter- and intra-basin variabilities of $\tau$ in CONUS (Fig. 10). The highest dependence is observed along the Northwest coast, where $\tau$ is approximately 0.2. The other coastal regions generally have a lower $\tau$ ($0\sim0.1$) and the $\tau$ value normally decreases upstream within the river basins. The intra-basin variability of $\tau$ is the greatest in the West coast.
Lastly, we compute $P_{Q,SS}$ for the river basins along the CONUS coast (Fig. 11). The value of $P_{Q,SS}$ shows larger variabilities than that of $P_Q$, $P_{ss}$ and $\tau$ as it includes uncertainties in the marginal distribution and dependence structure as well as uncertainty of the copula function selection. The highest $P_{Q,SS}$ ($\sim0.4\%$) is observed in the Northwest and Gulf coasts, while that in the East coast is generally less than 0.3%. The greater variability of $P_{Q,SS}$ in the West and Gulf coast basins is consistent with the spatial pattern of the marginal exceedance probability and dependence (Figs. 8~10). Interestingly, in
several basins (Fig. S8), $P_{Q,SS}$ differs significantly between the mainstem and tributaries.

The spatial variabilities of the CF hazard can be attributed to the inherent variability in both streamflow and storm surge. As implied by the spatially varied relative importance of $Q$, $SS$ and riverbed elevation (Fig. 4), hydrological patterns exhibit substantial variation among river basins depending on river topology, basin characteristics and other geomorphological factors. Storm surge is also influenced by local factors of coastal topography, bathymetry and storm characteristics. These combined factors result in varying frequencies, durations, and correlations of fluvial and coastal flooding. For instance, in the western Gulf coast, the rivers experience more frequent extremes (Fig. 8c); however, these extremes show low correlation with $SS$ in the same region, leading to a low dependence ($\tau$) (Fig. 10c). Similarly, the inter-basin variabilities depend on the spatial heterogeneity of basin characteristics. The timing and magnitudes of the streamflow peaks typically vary significantly between upstream gauges and the outlet (Fig. 6), and between smaller streams and major rivers, resulting in different values of $P_Q$ and $\tau$.

The CF hazard computed in this study shows both similarities and notable differences compared with previous statistics-based CFRAs (Eilander et al., 2020; Nasr et al., 2021; Wahl et al., 2015). For example, our analysis reveals several localized hotspots of the CF hazard characterized by a strong dependence between $Q$ and $SS$ in the Northwest and Gulf coasts (Fig. 12), as indicated by Eilander et al. (2020). However, the calculated $\tau$ values in our study are generally lower than those computed using annual maxima sampled from $Q$ and $SS$ observations in the East coast (Nasr et al., 2021). Also, our $\tau$ values are higher than that derived from the dependence of $Q$ and precipitation along the West coast, and a previous study also demonstrated substantial variations in $\tau$ at specific locations when using different sampling approaches for the two CF drivers (Wahl et al., 2015). These differences result from variations in the sampling of extreme events, the specific CF drivers considered, the statistical methods employed, as well as other uncertainty sources discussed in Section 2.2.1. Despite the variations observed among different frameworks, each study provides unique insights into the understanding and addresses the complexities associated with CF risks.

### 3.2.2 CF exposure

The cumulative population exposed to CF over the simulation period is computed to represent the CF exposure (Fig. 12). The human exposure to CF varies from 0 to 10,000 people and is restricted to the coastline. This is not unexpected because the CF-impacted riverine regions are governed by the river topology and the amplitude of $SS$ at the river outlets. Overall, this CF exposure is low because (a) the backwater extent is limited to the low-gradient regions, and (b) the occurrence of CF events is low over the 40-year period. Although the CF exposure for the West coast is only observed at the river outlets because of the large riverbed elevation impact (Fig. 7), it can extend several cells ($O(10^4$ m)) upstream in several river basins of the East and Gulf coasts. Also, the CF exposure has a spatial variability very different from the CF hazard, for example in the Northwest coast. These findings demonstrate the necessity to account for the impacts of river topology and calculate spatially distributed risks in CFRA.

### 3.2.3 CF risk

The CF risk is derived based on the CF hazard and exposure (Fig. 13). The CF risk varies significantly along the CONUS coast and is the highest at the river outlets although the risk can be present to a much larger extent over most river basins of the East and Gulf coasts due to the upstream propagation of backwaters. This pattern is consistent with the flood exposure. In a few low-lying regions of the East and Gulf coasts, the CF risk extends several cells upstream from the river-ocean interface. In addition, we identify a few hotspots ranked using the *CFRI* averaged over a river basin (Table 3). Table 3 shows that the coastal area of San Joaquin River and Hudson River where Silicon Valley and New York City are located, respectively, are particularly vulnerable to CF. The total exposed population and the maximum exceedance probability ($P_{Q,SS}$) are also provided. Although the three metrics correspond to different types of CF risks, these hotspots require extra attention in CF management. They also provide target regions where the computational mesh should be refined to improve model accuracy. Overall, the CF risk accounts for the occurrence rate of CF events, the impacts of basin characteristics and population density. The proposed CFRA avoids the biased risk estimation made by either statistics-based or hydrodynamics-based CFRA alone and can capture the minimum risk from their respective aspect.

## 4 Discussions

### 4.1 Differences between CFRAs

We examined the CF risk along the CONUS coast using different approaches based on the co-occurrence probability of a fluvial flood and a storm surge event, the human exposure to the CF events, as well as their combined impacts. The comparison shows that the different CFRA approaches result in significantly different CF risk estimates. The difference is remarkable in a few key regions. For example, in the San Francisco Bay area, while the CF hazard is low, the CF exposure is high due to the dense population and flat topology. In contrast, although the lower Mississippi River basin is endangered by backwater flooding, the probability of a CF event is low over the 40-year period. The difference in the estimated risks could be partially explained by whether more detailed physical processes and topological factors are considered. However, the overall CF risk is also valuable. For example, we find that although the Northwest coast (e.g., Puget Sound) has a high CF hazard, such risk is only restricted to the coastline and does not extend to the upstream regions. In summary, to comprehensively understand the complex CF risk, it is critical for CFRAs to integrate multiple risk factors based on the available approaches.

The proposed CFRA also draws attention to the CF risk in upstream river basins which are usually ignored in the large-scale statistics-based CFRA. Typically, the flood risks related to coastal hazards (e.g., storm surge) are limited to the land-ocean interface. However, through the river networks, the backwater effects can propagate upstream by hundreds of kilometers in low-lying watersheds (Lamb et al., 2012). Our results show the CF risks extending upstream over several river basins (Fig. 12 and 13).

CF hazard and exposure can also be impacted by climate change as CF drivers interact with climate drivers (Zscheischler et al., 2020). Global warming will likely increase the frequency of extreme precipitation (Alfieri et al., 2016), the intensity of river discharge (Bermúdez et al., 2021) and storm surge (Camelo et al., 2020), and the duration of the fluvial and coastal flooding (Feng et al., 2022) in many regions, such as the US east coast (Ting et al., 2019). All these factors contribute to the exacerbation of CF risks, as both marginal and joint exceedance probabilities will increase. Moreover, climate change has the potential to alter the characteristics or distributions of CF drivers. For instance, the dependence between storm surge and precipitation is enhanced by climate warming, which increases the CF hazard (Wahl et al., 2015). The elevated sea level will move the backwater extent further upstream, increasing the CF exposure (Kulp and Strauss, 2019). Given the uncertainty of climate change, more attention should be paid to understand the potential impacts of different socioeconomic pathways on the CF risk.

## 4.2 Limitations and Future work

The CFRA framework provides an effective tool to support large-scale CF risk management in CONUS. However, this study has a few limitations that warrant further improvements. First, the 40-year time series is relatively short for deriving robust extreme statistics, as extreme events can have much longer return periods (Apel et al., 2004). This type of epistemic uncertainty can be reduced by using data that covers a longer period. The simulation period is limited by the available large-scale runoff forcing (Yang et al., 2021) and the GTSM reanalysis dataset (Muis et al., 2022). The reanalysis forcing typically covers a shorter period (e.g. from 1979 to 2018) and thus statistical analyses based on such periods may lead to unreliable estimates of the return frequency/probability of TCs. While it remains challenging to determine the sufficient data length, such uncertainty likely depends on the region-specific exceedance probability, particularly when lower probabilities correspond to longer return periods. ESMs, which can simulate CF drivers for longer periods than reanalysis data, can be used to analyze historical CF risks. Moreover, high-resolution cloud-resolved ESMs show promising performance in representing extreme events (Caldwell et al., 2021), which can help the hydrodynamics-based CFRA quantification. The ability to better represent the complex processes at the terrestrial and aquatic interface, could also increase the simulation accuracy for extreme events. For example, several related new capabilities have recently been developed for Energy Exascale Earth System Model (E3SM) (Golaz et al., 2022), such as the multi-scale variable-resolution meshes and state-of-the-art techniques developed in E3SM land and ocean models (Lilly et al., 2023; Pal et al., 2023). These advancements have shown the potential to improve the representation of climate extremes, which is important for the CF risk assessment.

Second, we quantify the CF impacts using the *SS*-induced backwater effects without considering the complex interactions between the two flood drivers or the possibility of a CF event induced by an individual extreme driver. CF does not necessarily require all drivers to exceed their corresponding thresholds (Zscheischler et al., 2020). The backwater effects are driven in MOSART by prescribing the *SS* time series at the downstream boundary. However, the actual CF is driven by the interactive processes between multiple drivers, including precipitation, land surface runoff and inundation, river discharge

and coastal tide, storm surge and wave (Nasr et al., 2021). For example, the interaction between flooded water and channel flow, groundwater and surface water, river discharge and upstream propagation of ocean tides and storm surge will likely attenuate the hydrograph, intensify inland flooding or affect the backwater propagation, particularly in low-lying watersheds. Such interactions contribute to another layer of complexity and uncertainty at the terrestrial and aquatic interface and should be simulated using ESMs with fully coupled land, river and ocean modeling components. Such interactions contribute to another layer of complexity and uncertainty at the terrestrial and aquatic interface and should be simulated using ESMs with fully coupled land, river and ocean modeling components. Interactive coupling has been developed within the Energy Exascale Earth System Model (E3SM) (Feng et al., 2022; D. Xu et al., 2022) for further CFRA developments. Third, the MOSART simulated floodplain inundation could be a sensitive factor when estimating the CF exposure, as MOSART employs a macro-scale inundation scheme (Luo et al., 2017) and the model has limited resolution for evaluating risk in high population density areas. Validation of the floodplain inundation over coastal river basins is challenging because the CF inundation data is limited and such data does not differentiate coastal inundation and river floodplain inundation. Last but not least, the uncertainty sources identified in this study are undoubtedly only "the tip of the iceberg". There are many other uncertainties related to parameterization and structural errors of data and physical models. For example, the 1/8° MOSART grid is appropriate for continental-scale multi-decadal simulations. The mesh resolution is insufficient to resolve the distributed risk within a grid cell, as neither human residence nor topology can be resolved to represent the flood exposure. The input data, such as surface runoff and digital elevation models (DEMs), have uncertainties that should be quantified. This is a well-acknowledged challenge in large-scale modeling (Cook & Merwade, 2009; Van de Sande et al., 2012). As quantifying the uncertainty of simulated runoff and streamflow in ESMs remains challenging (Lawrence et al., 2019), fully addressing such uncertainty in CFRAs is beyond the scope of this study.

## 5 Conclusion

This research proposes a CFRA framework to investigate the CF risk along the CONUS coast. This framework includes both statistics-based and hydrodynamics-based CFRAs and assesses the CF hazard, exposure and the overall risk using a bivariate statistical model of river discharge and storm surge and the large-scale MOSART river model coupled with the global GTSM reanalysis dataset. The resulting CF risks show substantial variabilities at the inter- and intra-basin scales. In particular, the variability is significant in the CF hazard and along the West and Gulf coasts. More importantly, the three risk measures show very different spatial patterns and hotspots depending on the local settings. The high occurrence probability of a CF event does not necessarily pose high CF exposure. Thus, it is important to understand the different risk types and avoid biased risk estimation using either statistics-based or hydrodynamics-based assessment methods singly. Using the new CF risk index, we identify that the coastal area of San Joaquin River and Hudson River where Silicon Valley and New York City are located, respectively, are particularly vulnerable to CF.

Moreover, we identify the uncertainty sources in the existing CFRAs. Even though statistics-based CFRA is widely used in continental and global domains, the estimated CF risks based on such CFRAs could be too high because flow physics, such as complex fluvial processes and the backwater propagation, are neglected. The hydrodynamics-based CFRA is more appropriate for analyzing the spatially-distributed risk but the large-scale numerical models likely underestimate the flood extremes and the dependence structure among the CF drivers. A more robust CFRA requires improved performance in large-scale modeling. In the future, we plan to apply the land-river-ocean fully coupled E3SM on a coastal-refined mesh to better represent the interactive CF physics and develop a more robust CFRA.

**Code and data availability**

The water level observation along the CONUS coastline is obtained from NOAA tides & currents website ( https://tidesandcurrents.noaa.gov/) (NOAA, 2022). The streamflow measurements are downloaded from USGS National Water Information System (NWIS) website (https://waterdata.usgs.gov/nwis) (U.S. Geological Survey, 2016). The GTSM storm surge ($SS$) simulation is available from the Copernicus Climate Change Service (C3S) Climate Data Store (Yan et al., 2022). The MOSART source code, the statistical analysis code of the compound flood risk assessment, the MOSART simulation output and the statistical analysis results are available on Zenodo (https://doi.org/10.5281/zenodo.7588256, Feng, 2023).

**Author contributions**

DF and ZT devised the framework of the compound flood risk assessment and designed the statistical and numerical analyses. DX created the MOSART runoff forcing from the GRFR dataset. DF carried out the model simulation, risk analysis and visualization. All authors discussed and reviewed the analysis and results and contributed to the manuscript editing. ZT and LL supervised the project.

**Competing interests**

The authors declare that they have no conflict of interest.

**Acknowledgements**

The research presented in this manuscript is supported by the Earth System Model Development program areas of the U.S. Department of Energy, Office of Science, Office of Biological and Environmental Research as part of the multi-program,

collaborative Integrated Coastal Modeling (ICoM) project. PNNL is operated for DOE by Battelle Memorial Institute,
United States under contract DE-AC05-76RL01830. All model simulations were performed using resources available
through Research Computing at Pacific Northwest National Laboratory.

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

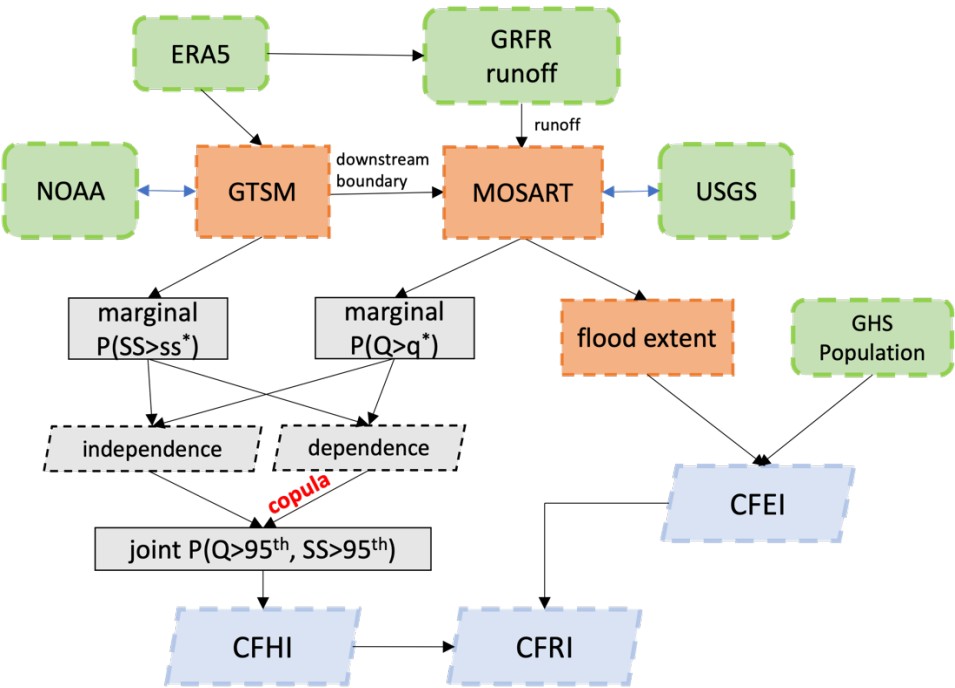

**Figure 1: The CFRA framework. Data, numerical modeling, statistical modeling, and risk calculation are represented by green, orange, gray, and blue colors, respectively.**

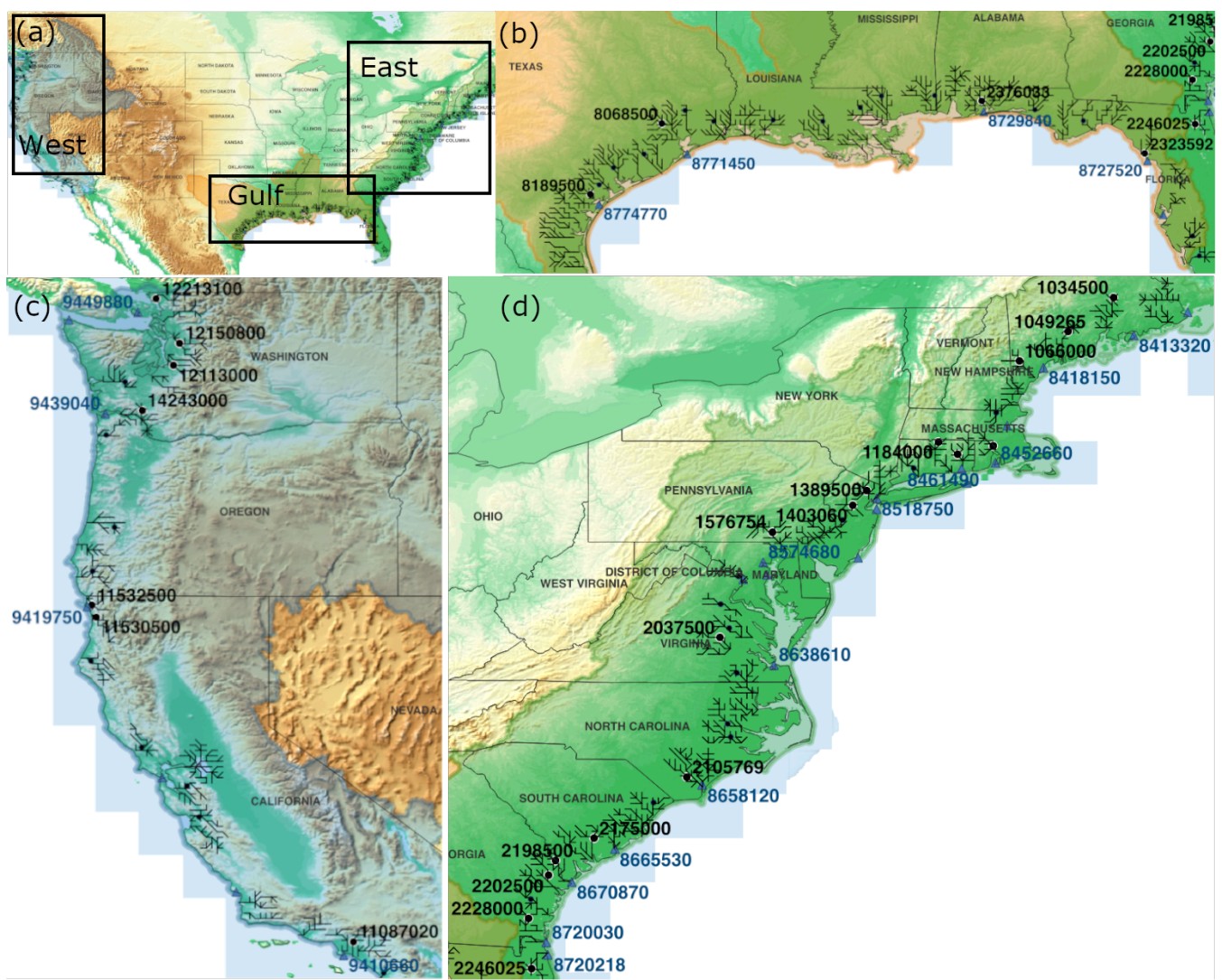

**Figure 2: Study domain and observations overlaid on the USGS 3D elevation (U.S. Geological Survey, 2019). The black and blue circles represent the USGS and NOAA gauges, respectively. The gauges used for identifying uncertainties are labeled with the gauge ID. The black solid lines are the coastal river network that consists of at most seven cells from each river outlet.**

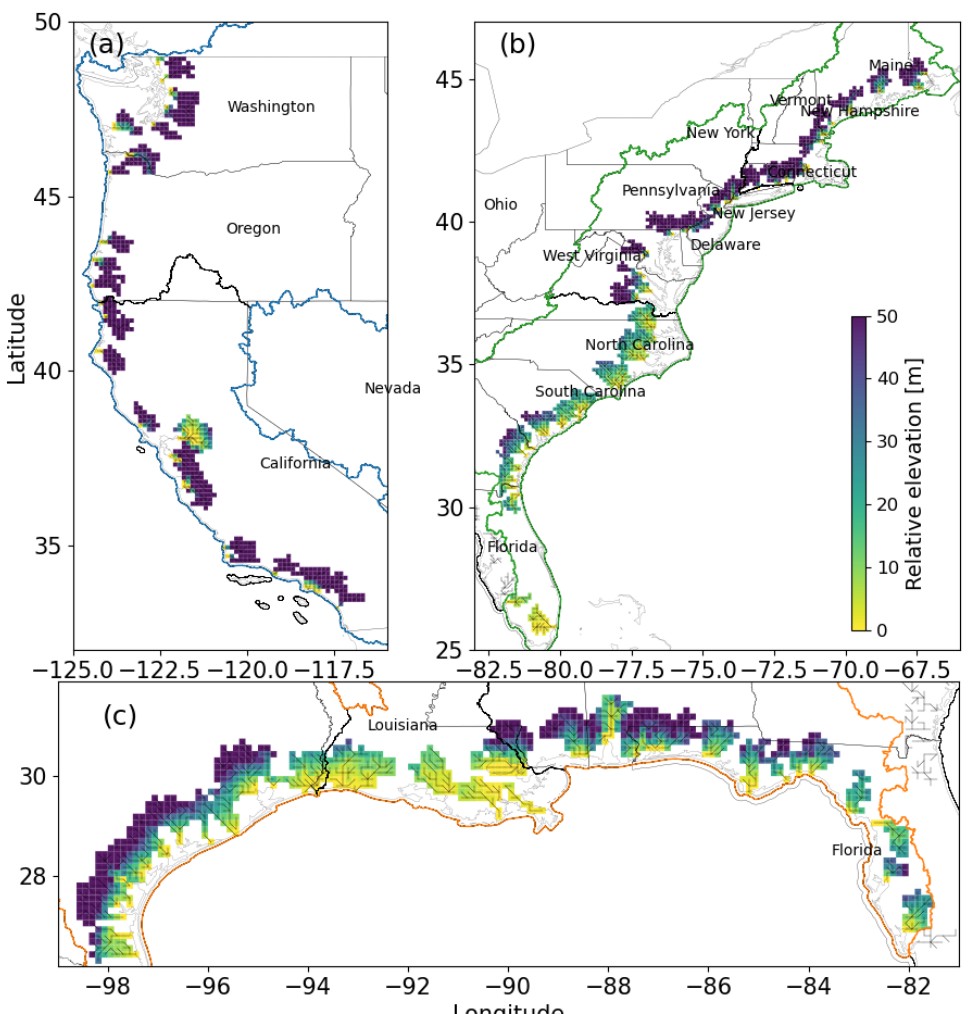

**Figure 3: The relative riverbed elevation along the (a) West coast, (b) East coast and (c) Gulf coast. The river networks within the MOSART coastal cells are shown as black solid lines.**

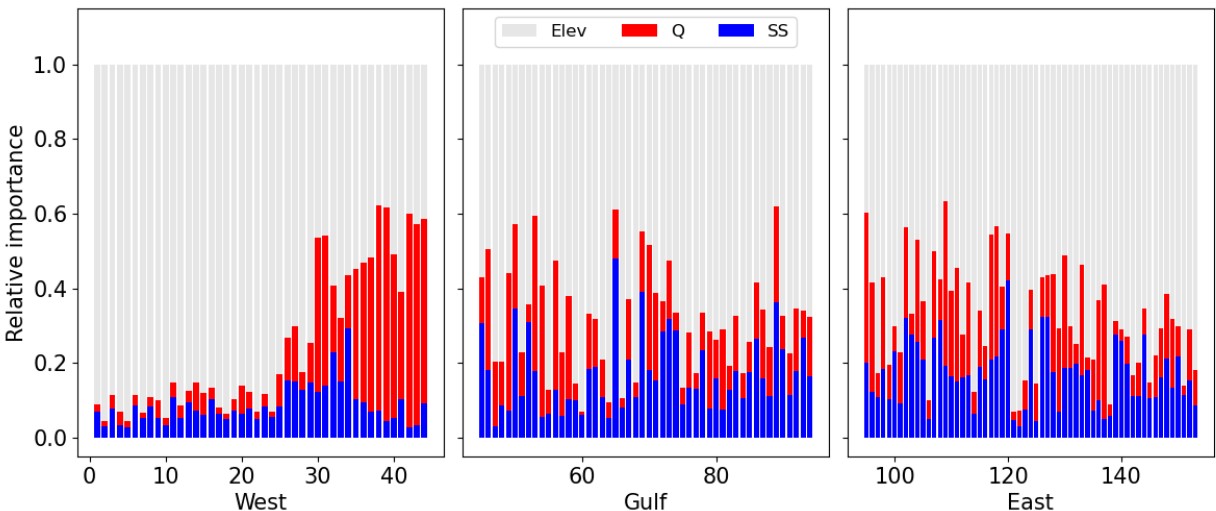

**Figure 4: The relative importance of $Q$ (red), $SS$ (blue) and relative riverbed elevation (gray) provided in the counter-clockwise order of the river basins along the West, Gulf and East coastlines. The numbers representing individual river basins correspond to those in Figure S3.**


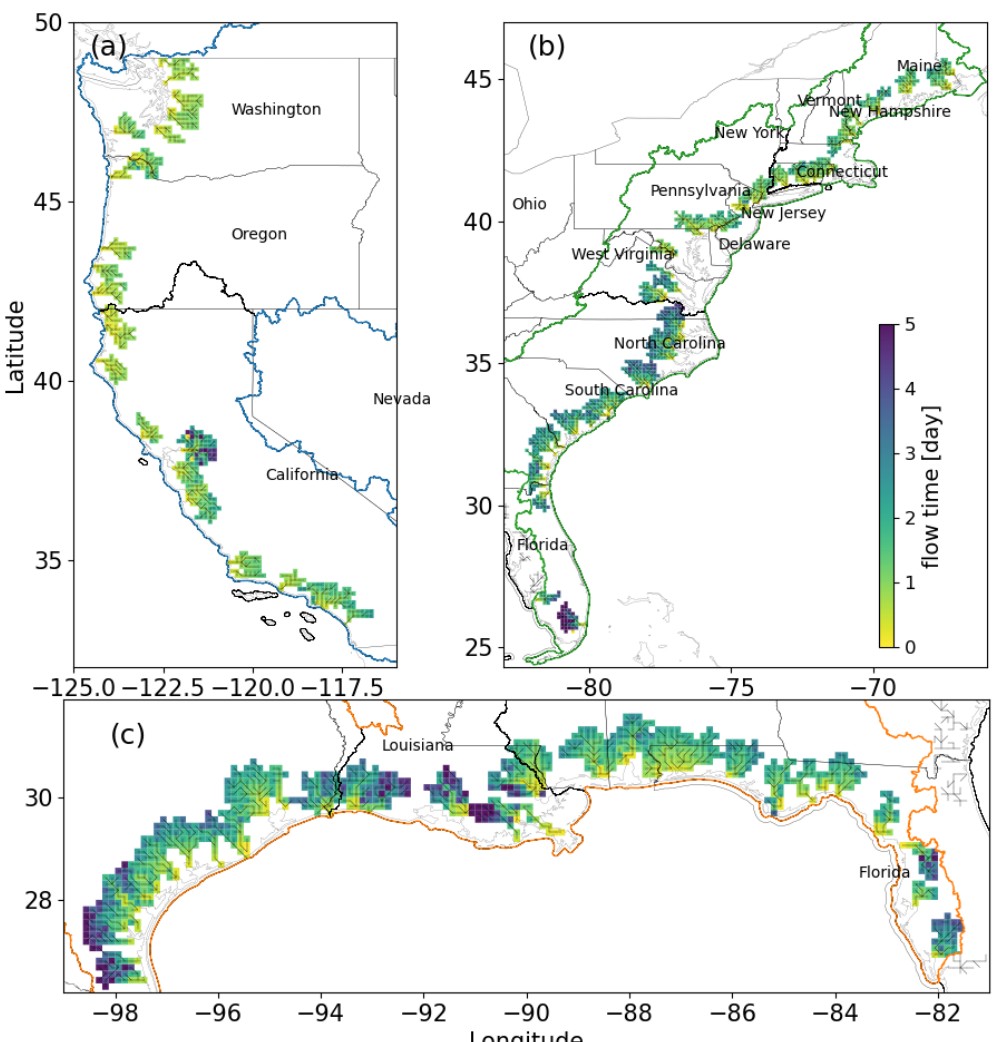

**Figure 5: The flow time of river discharge at the MOSART coastal cells along the (a) West coast, (b) East coast and (c) Gulf coast. The river networks within the MOSART coastal cells are shown as black solid lines.**

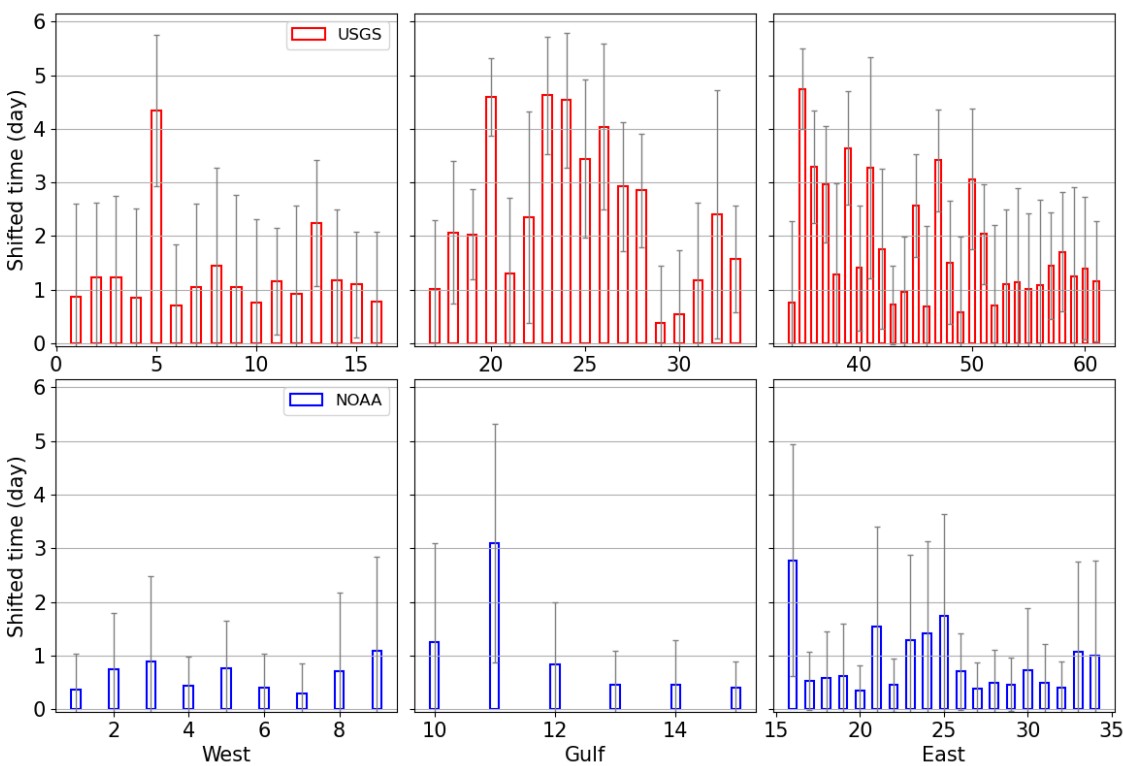

**Figure 6: The shifted days in the *Q* and *SS* peaks between the observation gauges and the corresponding river outlets provided in the counter-clockwise order of gauges along the West, Gulf and East coastlines. The rectangular box represents the averaged shift over the simulation period and the error bar represents the standard deviation. The numbers representing observation gauges correspond to those in Figure S4.**

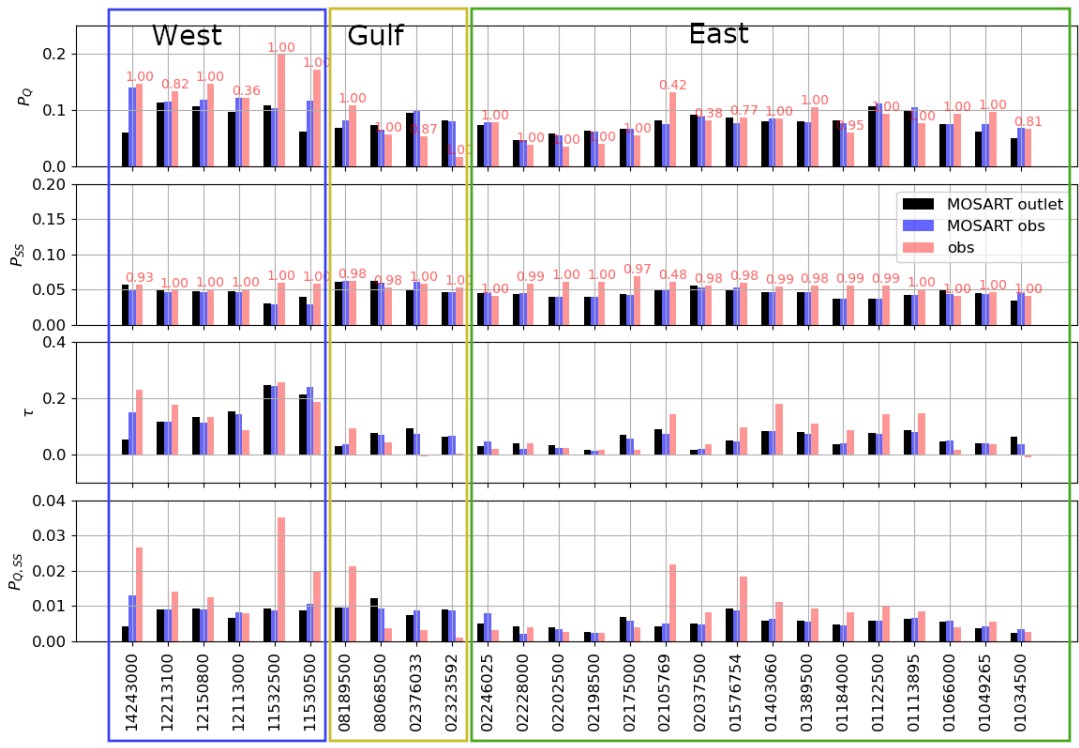


**Figure 7: The comparison of $P_Q$, $P_{SS}$, $\tau$ and $P_{Q,SS}$ computed from the modeled $Q$ and $SS$ at the river outlet (black), modeled $Q$ and $SS$ at the observation gauges (blue), measured $Q$ and $SS$ at the observation gauges (pink). The number on top of each bar is the percentage of the data coverage.**

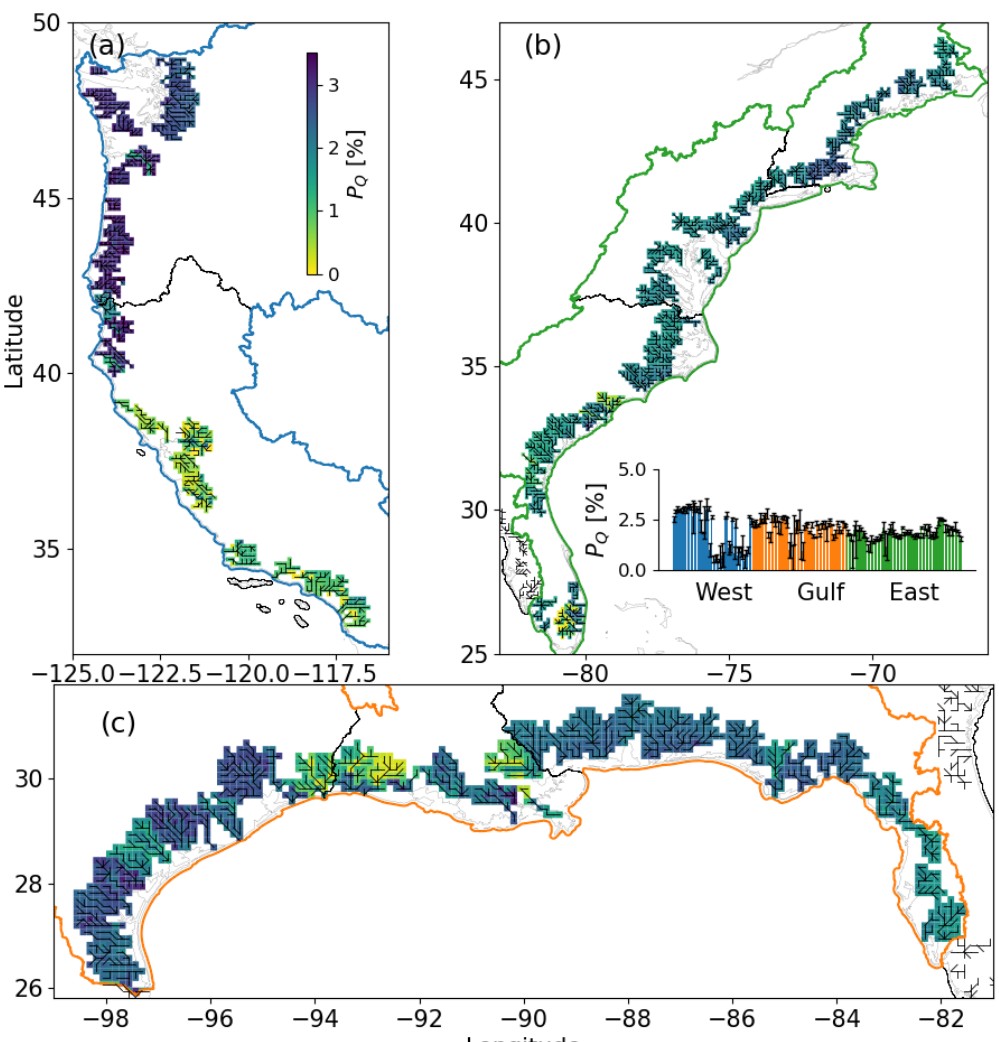

**Figure 8: The marginal exceedance probability $P_Q$ along the (a) West coast, (b) East coast and (c) Gulf coast. The basin-averaged $P_Q$ is provided in the counter-clockwise order of the basins along the US coast in the lower left insert of subplot (b) where the error bars represent the corresponding standard deviation.**

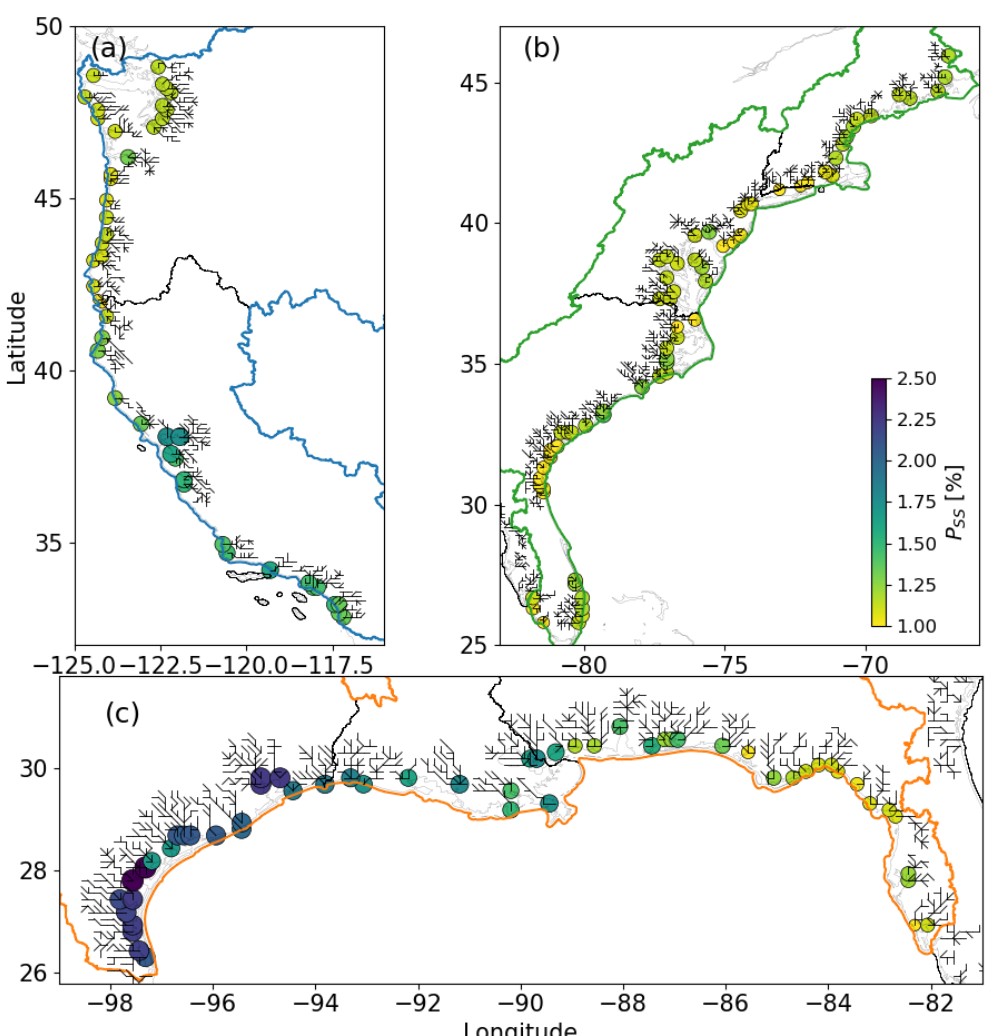

**Figure 9: The marginal exceedance probability $P_{SS}$ located at the GTSM cells nearest to the corresponding outlets.**


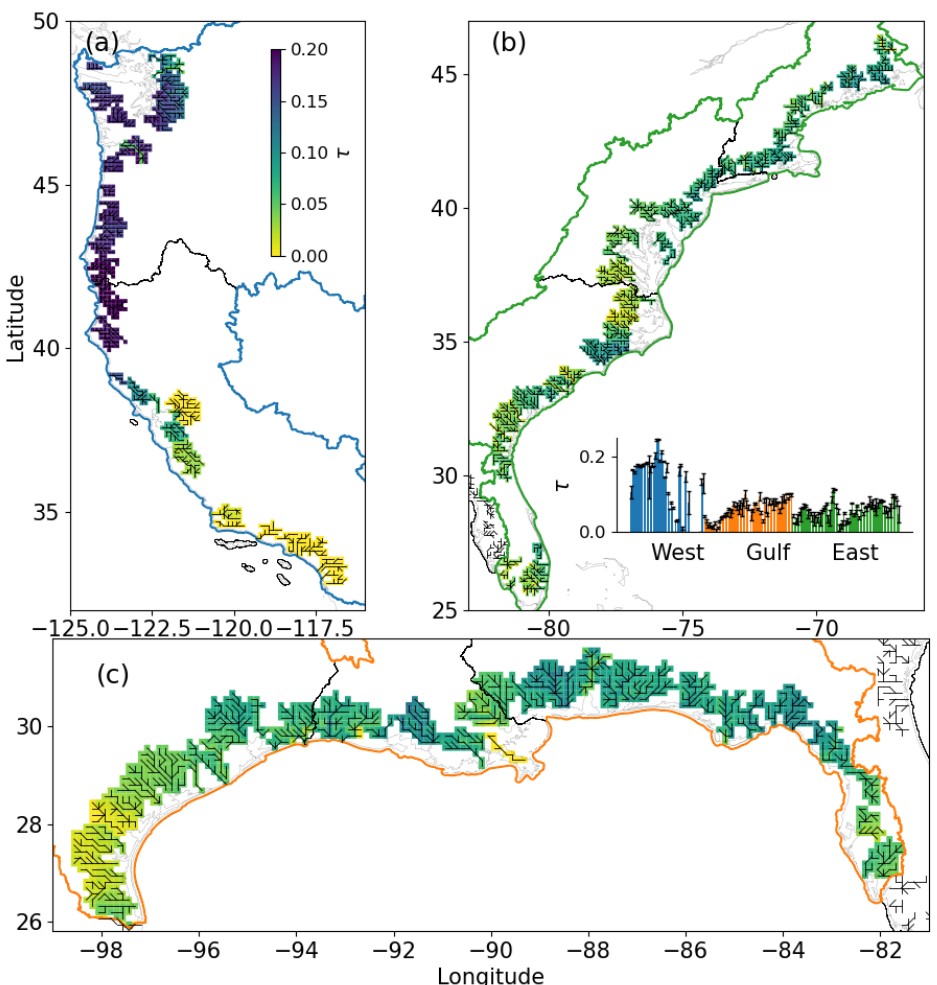

**Figure 10: The Kendall's correlation coefficient ($\tau$) computed for each MOSART coastal cell using the corresponding $Q$ and $SS$ (Section 2.1). The insert in subplot (b) illustrates the basin-averaged value of $\tau$ provided in the counter-clockwise order of the river basins along the West, Gulf and East coastlines with the error bars representing the corresponding standard deviation.**

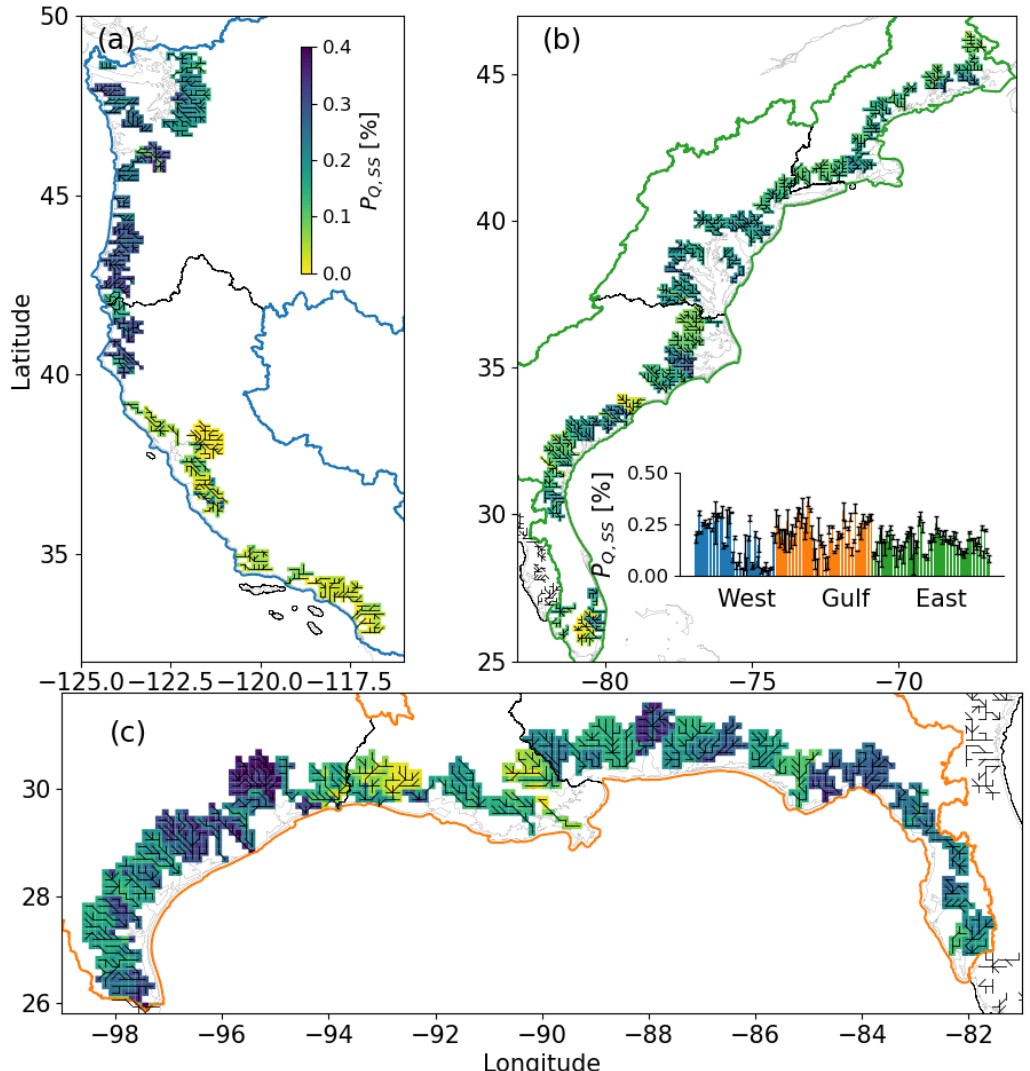


**Figure 11: The joint exceedance probability ($P_{Q,SS}$) computed for each MOSART coastal cell using Eq. 3 (Section 2.1). The insert in subplot (b) is the basin-averaged value of $P_{Q,SS}$ provided in the counter-clockwise order of the river basins along the West, Gulf and East coastlines with each error bar representing the corresponding standard deviation.**

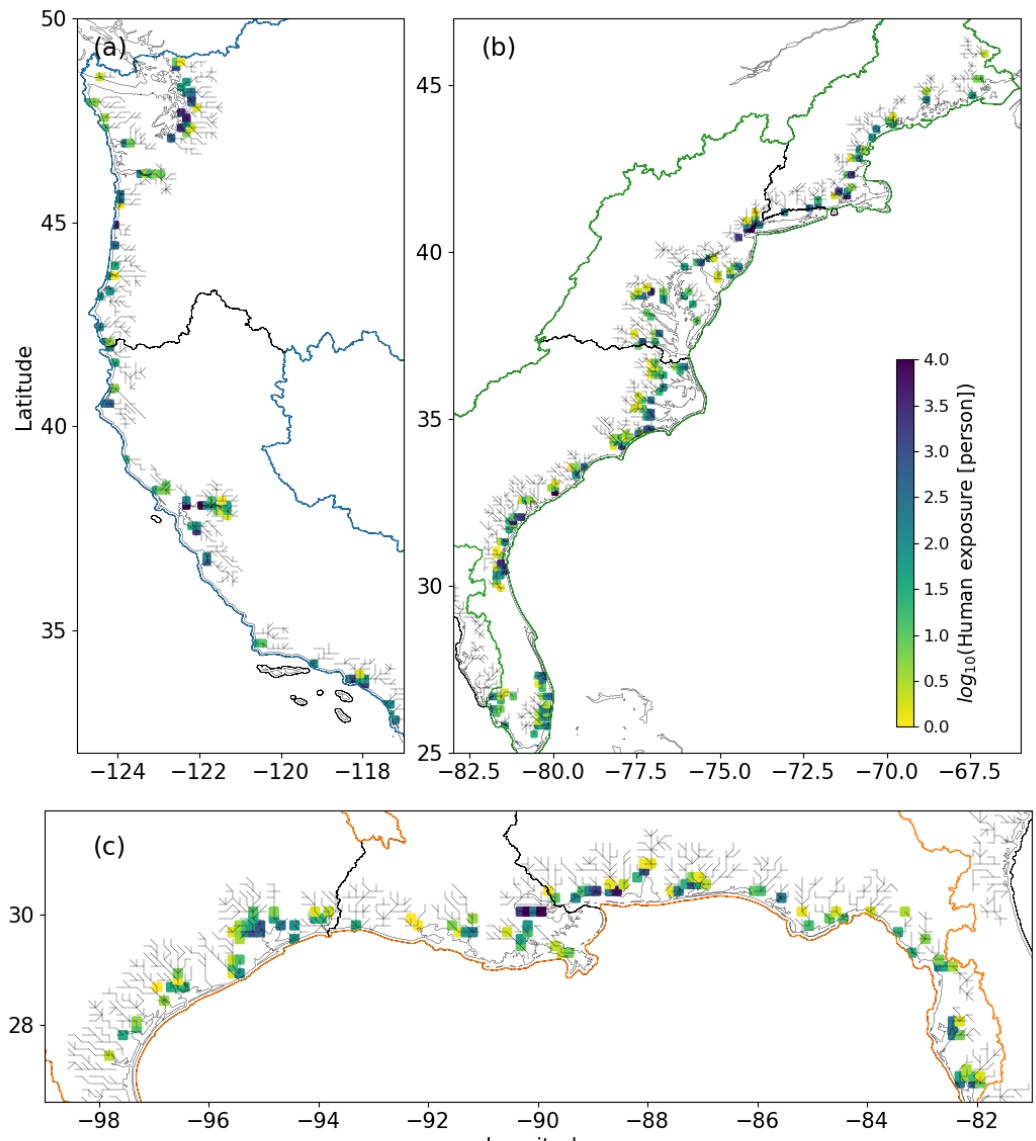

Figure 12: The accumulated population exposed to CF over the simulation period.

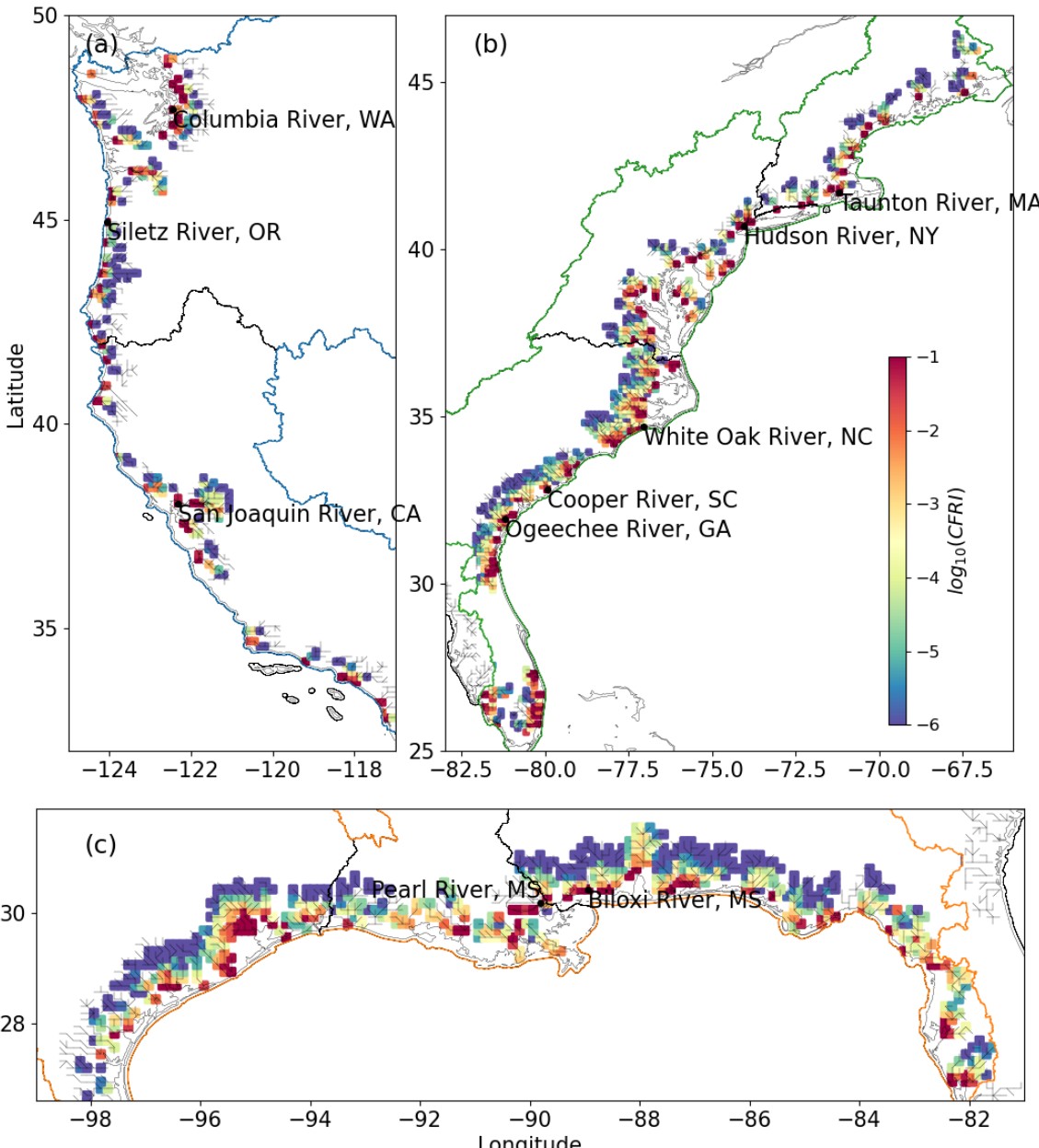

**Figure 13: The *CFRI* represented on a logarithmic scale using a base of 10 along the (a) West coast, (b) East coast and (c) Gulf coast.**


Table 1: Uncertainty sources in CFRA. Sources in boldfaces are considered in the analysis.

| Module | Aleatory uncertainty | Epistemic uncertainty |
|---|---|---|
| **Statistical models** | **Spatial variability in fluvial processes and river topology** | **Measurement uncertainty (e.g., measurement errors, inappropriate in-situ locations, and limited data coverage or partial time series)** |
| | Non-stationarity | Model structure uncertainty (e.g., selection of probability distribution functions and selection of dependence models) |
| **Numerical models** | Future climate change | Parameter uncertainty |
| | | Data uncertainty (e.g., uncertain river topology and channel geometry data) |
| | | Model structure uncertainty (e.g., simplified flood wave physics, uncertain runoff generation schemes, and coarse spatial resolutions) |

Table 2: Three combinations of $Q$ and $SS$ for model-data comparison.

| | $Q$ | $SS$ |
|---|---|---|
| **a** | MOSART modeled value at the river outlet | GTSM modeled value at the river outlet |
| **b** | MOSART modeled value at the USGS gauge | GTSM modeled value at the NOAA gauge |
| **c** | USGS measurement | NOAA measurement |

Table 3: Top ten rivers with high CF risk. The locations are shown in Figure 13. CFRI represents the averaged CF comprehensive risk index (Eq (7)) in the river basin. Population exposure is the total population exposed to CF over the simulation period.

| No. | River name | River outlet Location | CFRI | Population exposure [person] | Maximum CF probability (%) |
|---|---|---|---|---|---|
| 1 | Ogeechee River, GA | 31.9375, -81.1875 | 38.617 | 4,617 | 0.158 |
| 2 | Cooper River, SC | 32.8125, -79.9375 | 36.468 | 11,096 | 0.264 |
| 3 | San Joaquin River, CA | 38.0625, -122.3125 | 33.182 | 19,621 | 0.075 |
| 4 | Pearl River, MS | 30.1875, -89.8125 | 19.829 | 103,314 | 0.131 |
| 5 | Hudson River, NY | 40.6875, -74.0625 | 19.516 | 22,542 | 0.151 |
| 6 | White Oak River, NC | 34.6875, -77.0625 | 19.076 | 1,849 | 0.283 |
| 7 | Biloxi River, MS | 30.4375, -88.9375 | 12.414 | 2,261 | 0.253 |
| 8 | Siletz River, OR | 44.9375, -124.0625 | 11.467 | 2,438 | 0.278 |
| 9 | Columbia River, WA | 47.6875, -122.4375 | 10.323 | 1,567 | 0.310 |
| 10 | Taunton River, MA | 41.6875, -71.1875 | 9.528 | 3,795 | 0.186 |