# Peer review of "Understanding the Compound Flood Risk along the Coast of the Contiguous United States"

_Hydrology and Earth System Sciences, 2023_

## Author Comment (AC1)

Response to Reviewers

Title:  Understanding the Compound Flood Risk along the Coast of the Contiguous United States

Author Response 1st revision

Reviewer 1
Reviewer Comments:

In this manuscript, the authors apply a compound flood risk assessment (CFRA) to the coastal regions of the contiguous United States. In contrast to several previous studies, they do not only perform a bivariate extremes analysis between storm surge and river streamflow (referred to in their manuscript as data-based CFRA), but they couple their statistical analysis to a large-scale river-routing model and population exposure information (referred to in their manuscript as physics-based CFRA). Their main finding appears to be that their metrics, such as the marginal and joint exceedance probabilities and Kendall's rank correlation, differ substantially depending on the type of CFRA. The authors use their results to argue that 'data-based' CFRA alone does not provide a holistic view of CFRA and provides a biased view, hence different types of CFRA need to be considered. Although my experience in this field is limited, I think that this contribution is welcome and useful, and I also found it well written. I do have several comments, though, that I hope will help to improve the manuscript.

Author Response:

We appreciate the reviewer for the insightful comments and suggestions. In the following we address your comments point by point. Our responses and all changes in the revised paper are marked in blue. If you have further questions or concerns, please let us know. Although HESS does not allow us to share our revised manuscript at this stage of the review process, we provide excerpts throughout the response to help illustrate the changes. We will provide the revised manuscript when invited.

R1C1:

'Data-based CFRA': I am not sure if this is a commonly used term, but it is ambiguous to me because 'physics-based CFRA' is also based on data. Would it be possible to find terminology that more clearly distinguished the two types of CFRA? Perhaps 'statistics-based' and 'dynamics-based' or so?

Author Response:

We appreciate the reviewer's suggestion and agree that "statistics-based" and "dynamics-based" are better terminology for the two types of CFRAs.

This is the first comprehensive study to distinguish CFRA into two different types, so there is no adopted definition yet in the existing literature. According to the reviewer's suggestion, we found that "statistics-based" CFRA properly pertains to approaches that rely on the statistical analysis and data interpretation to understand the CF risk, where data can be either from observations or numerical models. "Dynamics-based" refers to the methodology that focuses on the study of how systems change over time and how various factors interact to determine the risk patterns, which would be a more appropriate terminology than "physics-based".

We have changed "data-based" to "statistics-based" and "physics-based" to "dynamics-based" throughout the revised manuscript.

R1C2:

Figures: for the maps, color schemes are used that I doubt are color-blind friendly/perceptually uniform. Could the authors please check and modify their figures where necessary?

Author Response:

Thanks for the comments. The color scheme is changed to 'viridis' and 'spectral' in all spatial maps of the revised manuscript (Figure 3, 5, 6 and 8~13) to ensure color-blind accessibility.

R1C3:

L64-L92: I am missing an introduction of existing large-scale compound flood risk studies with similar aims, such as https://nhess.copernicus.org/articles/23/823/2023/. I would encourage the authors to include a paragraph on such studies and explain the similarities and differences with their own manuscript. The same comment applies to the Discussion section.

Author Response:

We appreciate the helpful comment to improve our manuscript.

This work is not only inspired by and but also builds upon many previous studies using either statistical approaches or numerical simulations to quantify the CF risk. Although these studies have already been acknowledged in our literature review, we agree with the reviewer that more thorough discussions should be provided in the introduction to compare the existing studies and in the discussion to better explain the difference of the resulting CF risks when using different approaches.

In the introduction of the revised manuscript, we first elaborated on the literature review of the statistical approaches with a brief comparison of the listed studies:

"At regional and global scales, statistics-based CFRAs consider the CF hazard as statistical dependence or co-occurrence rate of multiple flood drivers including discharge and surge (Moftakhari et al., 2017; Sadegh et al., 2018; Muñoz et al., 2020), precipitation and surge (Bevacqua et al., 2019), discharge, surge, and wave (Camus et al., 2021), etc. Statistics-based CFRAs perform statistical analysis using long-term data at paired gauges near the land-ocean interface. The data can be obtained either from large-scale numerical simulations (Eilander et al., 2020; Nasr et al., 2021) or gauge observations (Ward et al., 2018; Paprotny et al., 2020). Bivariate or multivariate analyses are performed to measure the CF hazard in terms of the joint occurrence of event extremes (Salvadori et al., 2007; Zscheischler et al., 2020). The CF hazard is determined either using the extreme dependence among multiple CF drivers or the likelihood of their joint occurrence. The dependence structure can be assessed from correlation and/or tail dependence coefficients (Wahl et al., 2015; Nasr et al., 2021). The co-occurrence rate may be calculated as the joint exceedance probability when a single or multiple drivers are above their predefined thresholds

(Moftakhari et al., 2017), e.g., 95th or 99th percentile (Kew et al., 2013), which are defined as "OR" and "AND" hazard scenarios by Salvadori et al., 2016."

We elaborated on the existing studies of large-scale models and acknowledged a particular contribution from the large-scale coupled river-coastal models.

"Such models offer the capability to evaluate spatiotemporally varied CF drivers, flood extent and population exposure to CF events from basin to global scales over multiple decades (Ikeuchi et al., 2017; Eilander et al., 2020; Eilander et al., 2023)."

The existing CFRA studies in the CONUS domain was briefly discussed in the last paragraph of the introduction. We expanded this part with more details added.

"The contiguous United States (CONUS) (Fig. 1) consists of 48 states, with coastal counties occupying about 10% of the total area. There are 17 major port cities, and ~40% of the US population residing in coastal counties are subject to high coastal flooding risks (Hanson et al., 2011). A high-resolution analysis study including pluvial, fluvial, and coastal flooding, projected a significant changing pattern of the flood risk in CONUS under future climate scenarios (Bates et al., 2021). Particularly, the CF risk was previously evaluated for the CONUS coastline or major US coastal cities using statistics-based CFRAs in terms of the dependence between storm surge and precipitation (Wahl et al., 2015), seasonable dependence among multiple CF drivers (Nasr et al., 2021), and the joint probability in "OR" hazard scenarios in response to sea level rise (Moftakhari et al., 2017)."

Finally in the result discussion (the end of Section 3.2.1), we provided a comparison of the risk assessments between ours and other statistics-based CFRAs. We highlighted that the CF risk could vary significantly when considering different CF drivers or using different statistical approaches.

"The CF hazard computed in this study shows both similarities and notable differences compared with previous statistics-based CFRAs (Eilander et al., 2020; Nasr et al., 2021; Wahl et al., 2015). For example, our analysis reveals several localized hotspots of the CF hazard characterized by a strong dependence between $Q$ and $SS$ in the Northwest and Gulf coasts (Fig. 12), as indicated by Eilander et al. (2020). However, the calculated $\tau$ values in our study are generally lower than those computed using annual maxima sampled from $Q$ and $SS$ observations in the East coast (Nasr et al., 2021). Also, our $\tau$ values are higher than that derived from the dependence of $Q$ and precipitation along the West coast, and a previous study also demonstrated substantial variations in $\tau$ at specific locations when using different sampling approaches for the two CF drivers (Wahl et al., 2015). These differences result from variations in the sampling of extreme events, the specific CF drivers considered, the statistical methods employed, as well as other uncertainty sources discussed in Section 2.2.1. Despite the variations observed among different frameworks, each study provides unique insights into the understanding and addresses the complexities associated with CF risks. The choice of a specific CFRA depends on the local characteristics of the study area and the specific requirements of local flood planning and management."

R1C4:

L144: Kendall's rank correlation is computed, but it is not clear to me if/what lag is allowed between the two variables. Could the authors please also discuss how the samples for which the rank correlation is

computed are conditioned, i.e., are cases in which one of the variates but not necessarily both variables are extreme (two-sided) also considered or not, and why (not)?

Author Response:

We apologize for the typo and the missing information regarding this comment.

The first step in calculating the CF hazard should be referred as "CF event selection" instead of "storm surge event selection". In sampling the CF events, we first extracted $SS$ events using the event selection scheme proposed in Feng et al., 2022. This scheme uses a peak detection algorithm to filter independent $SS$ events. For each selected $SS$ event, a CF event is identified if $Q$ during the event is over the predefined threshold of 95th percentile. The advantage of this method is that it allows a more realistic representation of a $SS$ event than defining a time window around the peak, as $SS$ may last longer than a few days. We acknowledge that the duration of a fluvial flood event may not precisely align with the duration of $SS$ during a CF event. It is possible there is only a small overlap between the two events. But we don't allow any time lag between $Q$ and SS because the CF exposure is measured by the extent of backwater. Such effects are dominated by $SS$ and will only be significant during the selected $SS$ events.

The description of step (a) in the CF hazard calculation has been elaborated more clearly.

"(a) CF event selection: use a $SS$ event selection scheme (Feng et al., 2022) to extract all $SS$ events with the $SS$ level over 95th percentile and then in the selected SS events identify them as CF events if river discharge of the corresponding station during these events is also over 95th percentile;"

"As the first step, our event selection scheme samples independent $SS$ events from the time series data, which avoids dependence in the extremes and eliminates the need for declustering. We assume that the $Q$ extreme within each $SS$ event is independent as both frequency and duration of $SS$ are generally much smaller than that of fluvial flooding. While it is possible that the duration of a fluvial flood event does not precisely align with a $SS$ event, we don't include time lag between $Q$ and $SS$ in the consideration as this study specifically quantifies the CF impact based on the $SS$-driven backwater effects."

While recognizing that a CF event does not necessarily require all flood drivers to exceed their individual thresholds (Zscheischler et al., 2020), we don't consider the case when only one of the drivers is significant because it will otherwise be difficult to identify the CF events in a large-scale analysis. We appreciate the reviewer's remark. This limitation is discussed in Section 4.2 of the revised manuscript.

"Second, we quantify the CF impacts using the $SS$-induced backwater effects without considering the complex interactions between the two flood drivers or the possibility when a CF event is induced by an individual extreme driver. CF does not necessarily require all drivers to exceed their corresponding thresholds (Zscheischler et al., 2020)."

R1C5:

Section 2.1: information about declustering of the peak events seems to be missing, although declustering is necessary to avoid dependence in the extremes. Could the authors please explain if/how their data was declustered?

Author Response:

Thanks for the comment. According to R1C4, we applied a new event selection scheme to extract all $SS$ events. This method ensures that the selected $SS$ events are independent as the sign of water level is always changed even between two adjacent events. While this method does not ensure the independence of $Q$, the duration and frequency of a $SS$ event are generally much shorter than that of fluvial flooding. We added the explanation to the revision:

"As the first step, our event selection scheme samples independent $SS$ events from the time series data, which avoids dependence in the extremes and eliminates the need for declustering. We assume that the $Q$ extreme within each $SS$ event is independent as both frequency and duration of $SS$ are generally much smaller than that of fluvial flooding. While it is possible that the duration of a fluvial flood event does not precisely align with a $SS$ event, we don't include time lag between $Q$ and $SS$ in the consideration as this study specifically quantifies the CF impact based on the $SS$-driven backwater effects."

R1C6:

L318-332: this paragraph discusses figures that are not part of the main manuscript. I would either include both the discussion and those figures in the manuscript or put both in the supplements, and only briefly refer to them. Its current form is inconvenient to read in my opinion.

Author Response:

Thanks for the suggestion. The discussion associated with the figures is moved to the supplement and is briefly referred as

"The model-data comparison is provided for a few example basins to demonstrate the various types of uncertainties (see Supplement for further details)."

R1C7:

Section 3.2.1: A comparison with the results of previous ('data-based') studies in this region (e.g., Wahl et al., 2015; Nasr et al., 2021) would provide useful context, but is missing. Could this be added?

Author Response:

Thanks for your comment. We added an additional paragraph at the end of Section 3.2.1 to discuss the comparison with previous studies. Please see the response to R1C3 for more details.

R1C8:

L400-402: while I appreciate this remark, data that covers a longer period cannot simply be obtained. Could the authors please explain/discuss how they imagine a dataset such as the GTSM reanalysis being extended back in time with reasonable confidence? In the absence of such data, how can we get a sense of how uncertain the results of the present study are given the relatively short period used? This seems

especially relevant given the underrepresentation of tropical cyclones in the observations/reanalysis data. And what about the influence of temporal variability on the dependence between the two variates examined? (e.g., Wahl et al., 2015)

Author Response:

We agree with the reviewer that the atmospheric reanalysis forcing usually covers a shorter period (e.g., from 1979 to 2018) and this causes epistemic uncertainty. This uncertainty is elaborated in the revision:

"The reanalysis forcing typically covers a shorter period (e.g. from 1979 to 2018) and thus may underestimate extreme events, such as TCs. While it remains challenging to determine the sufficient data length, such uncertainty likely depends on the region-specific exceedance probability, particularly when lower probabilities correspond to longer return periods."

The climate simulations and data used for climate research and assessments, including those from the Coupled Model Intercomparison Project Phase 6 (CMIP6), have uncertainties in representing extreme events. Despite substantial efforts, uncertainty quantification of the climate simulations remains challenging, which is beyond the scope of this study. We focused on the framework that assesses both CF hazard and exposure and highlight the uncertainties in the CF risk assessment. However, in the revision, we provide a promising path on how this uncertainty may be addressed in more advanced Earth system models. We briefly mentioned the new coupling capabilities developed within E3SM in Section 4.2:

"Such interactions contribute to another layer of complexity and uncertainty at the terrestrial and aquatic interface and should be simulated using ESMs with fully coupled land, river and ocean modeling components. Interactive coupling has been developed within the Energy Exascale Earth System Model (E3SM) (Feng et al., 2022; D. Xu et al., 2022) for further CFRA developments."

And in the revision, we provide other features that could contribute to more accurate climate simulations of extreme events:

"Moreover, the limited representation of extreme events may be addressed by more advanced ESMs. The ability to better represent the complex processes at the terrestrial and aquatic interface, could also increase the simulation accuracy for extreme events. For example, several related new capabilities have recently been developed for Energy Exascale Earth System Model (E3SM) (Golaz et al., 2022), such as the multi-scale variable-resolution meshes and state-of-the-art techniques developed in E3SM land and ocean models (Lilly et al., 2023; Pal et al., 2023). These advancements have shown the potential to improve the representation of climate extremes, which is important for the CF risk assessment."

We added a new paragraph to demonstrate the influence of climate variability:

"The broader definition of CF can be expanded to include the interaction between CF drivers and climate drivers (Zscheischler et al., 2020). Global warming will likely increase the frequency of extreme precipitation (Alfieri et al., 2016), the intensity of river discharge (Bermúdez et al., 2021) and storm surge (Camelo et al., 2020), and the duration of the fluvial and coastal flooding (Feng et al., 2022). All these factors contribute to the exacerbation of CF risks, as both marginal and joint exceedance probabilities will increase. Moreover, climate change has the potential to alter the characteristics or distributions of CF drivers. For instance, the dependence between storm surge and precipitation is enhanced by climate warming, which increases the CF hazard (Wahl et al., 2015). The elevated sea level will move the

backwater extent further upstream, increasing the CF exposure (Kulp and Strauss, 2019). Given the uncertainty of climate change, Earth System Models (ESMs) should be increasingly used to understand the potential impacts of different socioeconomic pathways on the CF risk."

R1C9:

L440-445: The open access to the code used for the paper is commendable. I strongly recommend doing the same for the output data instead of sharing it 'upon request'.

Author Response:

We apologize for the confusion. The MOSART simulation output and the statistical analysis results are already provided in this repository under the directory: statistical_analysis/files/. Because the size of the full MOSART simulation (including other variables in the whole CONUS domain) is too large, we saved the MOSART simulated streamflow for all coastal cells in the file below:

statistical_analysis/files/MOSART_discharge_GTSM_spatial.nc

We have revised the statement of the Code and data availability to ensure clarity:

"The MOSART source code, the statistical analysis code of the compound flood risk assessment, the MOSART simulation output and the statistical analysis results are available on Zenodo (https://doi.org/10.5281/zenodo.7588256, Feng, 2023)."

Minor issues:

R1C10:

L55: 'across rivers and estuaries': I suggest changing this to 'different rivers and estuaries', unless variation within a river or estuary is meant.

Author Response:

We have revised it as suggested.

R1C11:

L103 could the authors please comment on using a river routing model v.s. using a hydrological model that also includes groundwater? I am not an expert, but I can imagine that changes in groundwater also affect the propagation time between upstream and the coastal interface?

Author Response:

Thanks. The runoff used as input for MOSART is from a well validated hydrological model with groundwater processes (Yang et al., 2021). We agree with the reviewer that the impacts of groundwater and other river-land interactive processes contribute to more river dynamics. In response to this comment, we added more discussion on groundwater to Section 4.2:

"For example, the interaction between flooded water and channel flow, groundwater and surface water, river discharge and upstream propagation of ocean tides and storm surge will likely attenuate the hydrograph, intensify inland flooding or affect the backwater propagation, particularly in low-lying watersheds. Such interactions contribute to another layer of complexity and uncertainty at the terrestrial and aquatic interface and should be simulated using ESMs with fully coupled land, river and ocean modeling components."

R1C12:

L114: It only becomes clear later why the 2nd boundary condition is used (i.e., in eq. 6). It would be helpful to briefly motivate the 2nd boundary condition here already. Besides, it is not clear where this fixed level is derived from – is it the mean level in GTSM?

Author Response:

Thanks for the suggestion. We added a brief description to explain the use of simulations with the two BCs. Yes. The mean sea level is from GTSM. This sentence is rewritten to improve clarity:

"We apply two types of boundary condition (BC): (1) time-varying storm surge ($SS$) level and (2) fixed mean sea level. Both are obtained from the third-generation Global Tide and Surge Model (GTSM) (Muis et al., 2022). The $SS$-induced backwater effects in this study are quantified by comparing the two simulations which use the first and second BCs, respectively (Feng et al., 2022)."

R1C13:

L119: 'statistical model' please specify what this refers to.

Author Response:

We apologize for the confusion. We meant that the modeled streamflow used in the statistical analysis is from the MOSART simulation forced by the dynamic GTSM BC rather than the static MSL. In the revision, we have rewritten this sentence as below:

"The modeled streamflow used in the statistical analysis is from the MOSART simulation forced by the dynamic GTSM BC."

R1C14:

L175: 'using an R-package' please specify which package.

Author Response:

We use the R-package "copula" (Kojadinovic & Yan, 2010), which is now specified in the revised manuscript.

R1C15:

L192: is this necessary? Can the authors not just change the scale of the relevant plots?

Author Response:

The original intention of including this scaling factor is (a) to scale the risk index to 1~10, and (b) to increase the visibility of small values in the upstream regions. But as we plot $CFRI$ on the logarithmic scale using a base of 10, it is no longer necessary to upscale $CFRI$ using a factor. According to the reviewer's suggestion, we slightly changed the definition of $CFRI$ by excluding the factor of 10. We recomputed $CFRI$ and updated Figure 13 (original Figure 10).

"$CFRI = CFHI \times CFEI$, (7)

where the CF hazard index ($CFHI$) and the CF exposure index ($CFEI$) are calculated by normalizing $P_{Q,SS}$ and $W_p$ with their corresponding 95$^{th}$ percentile values."

[Figure]

**Figure 13: The *CFRI* represented on a logarithmic scale using a base of 10 along the (a) West coast, (b) East coast and (c) Gulf coast.**

R1C16:

L200: to avoid confusing this section for a results section, I suggest replacing 'investigate' with 'review' or 'discuss'. Additionally, I found this section not as well connected to the rest of the manuscript. Perhaps some sentences could be added as to why these uncertainty sources are introduced here.

Author Response:

Thanks for the comment. In the revision, we replaced "investigate" with "review" and added sentences to elaborate on the importance of uncertainty analysis and how such analysis is connected to CFRA.

"In this section, we review the uncertainty sources in the large-scale CFRAs. The uncertainty analysis is critical for a robust CFRA. Given that uncertainty can arise from diverse sources in both statistical and numerical models, an improved understanding of the uncertainty sources will provide valuable guidance for the future enhancement of the CFRA framework. We first examine the spatial variability in streamflow and storm surge and the relative impacts of riverbed elevations on the dynamics-based CFRA, as neglecting these physical factors leads to uncertainties in the statistics-based CFRA."

R1C17:

L206-207: would be more consistent to define the different types of uncertainty in the same order as they are introduced.

Author Response:

Thanks. This is corrected as suggested. "The uncertainty in CFRA can generally be classified into two categories: aleatory uncertainty and epistemic uncertainty."

R1C18:

L252: 'shift' is this the same as 'lag'? If so, I think that would be a clearer term.

Author Response:

We agree with the reviewer on this point and used the term "lag" when referring to the shift in streamflow peaks between an upstream USGS gauge and the river outlet.

"The lag in streamflow peaks between an upstream location and the river outlet will cause biases in the CFRA if the upstream Q measurements are used in the CF risk analysis."

"A longer flow time typically implies a larger time lag in streamflow peaks. In this study, the calculated flow time of each grid cell is averaged over the simulation period."

"The averaged time lag in the $Q$ peaks varies from 1 to 5 days, depending on the local topology, basin characteristics and hydrological response to coastal backwaters."

R1C19:

L258: I suggest to change +/- 5 days to: a window of 10 days around the extreme, if this is indeed the intended meaning.

Author Response:

Thanks. This is revised as suggested: "we then identify the peak date of each extreme event and define a time window of 10 days around the extreme".

R1C20:

Section 2.2.4: please consider putting this in a table, for a clearer overview.

Author Response:

Thanks. Table 2 is added for clarity: "We consider three combinations of $Q$ and $SS$ (Table 2):"

Table 2: Three combinations of $Q$ and $SS$ for model-data comparison.

|   | $Q$ | $SS$ |
|---|---|---|
| a | MOSART modeled value at the river outlet | GTSM modeled value at the river outlet |
| b | MOSART modeled value at the USGS gauge | GTSM modeled value at the NOAA gauge |
| c | USGS measurement | NOAA measurement |

R1C21:

L279: 'elevation gradient' – would it be worth plotting the gradient instead of the elevation itself in figure?

Author Response:

We apologize for the confusion and have replaced "elevation gradient" with "elevation" in the revision:

"In particular, the elevation effect dominates in the Northwest coast (Fig. 3), where the large riverbed elevation impedes the propagation of coastal backwaters and the areas of high CF risks are thus restricted to the coastline (Fig. 4)."

Based on our previous study (Fig. 10a in Feng et al., 2022), the extent of the backwater propagation is determined by the riverbed elevation. As we think the elevation may be more representative of the CF drivers, it is used in the random forest model to calculate the relative importance.

R1C22:

L286-287: could use some more explanation, in my opinion.

Author Response:

Thanks. More explanations have been added to this point.

"The averaged time lag in the $Q$ peaks varies from 1 to 5 days, depending on the local topology, basin characteristics and hydrological response to coastal backwaters."

R1C23:

L298: I think this needs to be 'Section 2.2.4 at 24 river basins', correct?

Author Response:

Thanks. Corrected.

Bermúdez, M., Farfán, J., Willems, P., and Cea, L.: Assessing the effects of climate change on compound flooding in coastal river areas, Water Resources Research, 57, e2020WR029321, 10.1029/2020WR029321, 2021.

Camelo, J., Mayo, T. L., and Gutmann, E. D.: Projected Climate Change Impacts on Hurricane Storm Surge Inundation in the Coastal United States, Frontiers in Built Environment, 207, 10.3389/fbuil.2020.588049, 2020.

Eilander, D., Couasnon, A., Leijnse, T., Ikeuchi, H., Yamazaki, D., Muis, S., Dullaart, J., Haag, A., Winsemius, H. C., and Ward, P. J.: A globally applicable framework for compound flood hazard modeling, Natural Hazards and Earth System Sciences, 23, 823-846, 2023.

Feng, D., Tan, Z., Engwirda, D., Liao, C., Xu, D., Bisht, G., Zhou, T., Li, H. Y., and Leung, L. R.: Investigating coastal backwater effects and flooding in the coastal zone using a global river transport model on an unstructured mesh, Hydrol. Earth Syst. Sci., 26, 5473-5491, 10.5194/hess-26-5473-2022, 2022.

Golaz, J. C., Van Roekel, L. P., Zheng, X., Roberts, A. F., Wolfe, J. D., Lin, W., Bradley, A. M., Tang, Q., Maltrud, M. E., and Forsyth, R. M.: The DOE E3SM Model Version 2: Overview of the physical model and initial model evaluation, Journal of Advances in Modeling Earth Systems, 14, 10.1029/2022ms003156, 2022.

Lilly, J. R., Capodaglio, G., Petersen, M. R., Brus, S. R., Engwirda, D., and Higdon, R. L.: Storm Surge Modeling as an Application of Local Time‐Stepping in MPAS‐Ocean, Journal of Advances in Modeling Earth Systems, 15, e2022MS003327, 10.1029/2022MS003327, 2023.

Kojadinovic, I. and Yan, J.: Modeling multivariate distributions with continuous margins using the copula R package, Journal of Statistical Software, 34, 1-20, 10.18637/jss.v034.i09, 2010.

Kulp, S. A. and Strauss, B. H.: New elevation data triple estimates of global vulnerability to sea-level rise and coastal flooding, Nature communications, 10, 1-12, s41467-019-12808-z, 2019.

Pal, N., Barton, K. N., Petersen, M. R., Brus, S. R., Engwirda, D., Arbic, B. K., Roberts, A. F., Westerink, J. J., and Wirasaet, D.: Barotropic tides in MPAS-Ocean (E3SM V2): impact of ice shelf cavities, Geoscientific Model Development, 16, 1297-1314, 10.5194/gmd-16-1297-2023, 2023.

Xu, D., Bisht, G., Zhou, T., Leung, L. R., and Pan, M.: Development of Land‐River Two‐Way Hydrologic Coupling for Floodplain Inundation in the Energy Exascale Earth System Model, Journal of Advances in Modeling Earth Systems, 14, e2021MS002772, 10.1029/2021MS002772, 2022.

Yang, Y., Pan, M., Lin, P., Beck, H. E., Zeng, Z., Yamazaki, D., David, C. H., Lu, H., Yang, K., and Hong, Y.: Global Reach-Level 3-Hourly River Flood Reanalysis (1980–2019), Bulletin of the American Meteorological Society, 102, E2086-E2105, 10.1175/BAMS-D-20-0057.1, 2021.

Zscheischler, J., Martius, O., Westra, S., Bevacqua, E., Raymond, C., Horton, R. M., van den Hurk, B., AghaKouchak, A., Jézéquel, A., and Mahecha, M. D.: A typology of compound weather and climate events, Nature reviews earth & environment, 1, 333-347, 10.1038/s43017-020-0060-z, 2020.

---

## Author Comment (AC2)

Response to Reviewers

Title: Understanding the Compound Flood Risk along the Coast of the Contiguous United States

Author Response 1st revision

Reviewer 2
Reviewer Comments:

This work has proposed a compound flood (CF) risk assessment (CFRA) framework for coastal regions in the contiguous United States (CONUS). Compound flood is a significant research topic, and this study is timely to contribute to expand the literature. Overall, this work reads smooth and has included great simulations and analysis. Below are some comments on the manuscript.

Author Response:

We would like to sincerely thank the reviewer for the valuable comments and recommendations. We have carefully addressed the reviewer's suggestions as follows, and we will provide the revised manuscript during the revision iteration.

R2C1:

1.     The scientific question was not quite clear. Is the objective to build a framework for assessing compound flood risk or to understand the risk across the U.S. coast? When reading the manuscript, it feels like the former was the main objective.

Author Response:

We appreciate the reviewer's comment. The major objective is to build the framework that can assess both compound flood hazard and exposure. Based on the framework, we provide an analysis to understand the risk across the US coasts. To better define the scientific question, in the introduction of the revision we first provide a general definition of CFRA before discussing details. Please note that we changed "data-based" and "physics-based" to "statistics-based" and "dynamics-based", respectively, according to the suggestion of another reviewer.

"Compound flood risk assessment (CFRA) is critical for flood planning, management, timely emergency response and decisions. CF risk has substantial spatial variabilities since the CF drivers and the CF risk dependence on the drivers are affected by the local conditions (Wahl et al., 2015), such as the characteristics of local basins that affect runoff generation, river routing (Hendry et al., 2019), synoptic weather systems, and storm characteristics (Seneviratne et al., 2012). CFRA can be classified into statistics-based and dynamics-based approaches. Statistics-based CFRAs rely on statistical modeling and define the CF hazard as the frequency of a CF event. Dynamics-based CFRAs use numerical simulations that can represent the spatiotemporal variabilities of CF drivers and how various CF drivers interact."

The objective is then defined at the end of the introduction:

"The objectives of this study are threefold: (a) to develop a new CFRA framework based on both statistical analyses and a large-scale river model that is coupled with a global ocean model reanalysis product; (b) to

provide a holistic hazard and exposure risk assessment of the compounding fluvial and coastal flooding along the CONUS coastline, and (c) to understand the uncertainties in both statistics-based and dynamics-based CFRAs."

R2C2:

2.   Line 160-165: "We consider $Q$ and $SS$ to be dependent of each other when they display a significant positive correlation (p-value<0.05)." Can we ignore the dependence if the p-value is smaller than 0.05. The authors need to better justify the selection of p-value.

Author Response:

Although 0.05 is a common choice, the p-value in this study is determined following a few previous studies in the CONUS domain (Ghanbari et al., 2021; Moftakhari et al., 2017). We justified the selection of p-value in the revision.

"The significance level is set as 0.05 following previous studies of statistics-based CFRAs (Ghanbari et al., 2021; Moftakhari et al., 2017)."

We agree that the choice of p-value is critical for determining dependence and would be a source of uncertainties in statistics-based CFRAs, which may be classified as model structure uncertainty (Table 1). This is now mentioned in Section 2.2.1.

"Moreover, model structure uncertainties always exist in statistical models, such as the selection of marginal distribution functions and dependence on level of significance."

R2C3:

3.   Line 170-175: Which copula did the authors choose to quantify the dependence and why?

Author Response:

Thanks for the comment. For each MOSART cell, we perform the same analysis to select copula from the same 24 candidates following the instructions provided by Moftakhari et al., 2017. Given the total number of grid cells, it is tedious to provide the copula name for each grid cell. But in the revised manuscript, we elaborated on the selection of copula:

"For each MOSART coastal cell where $Q$ and $SS$ are dependent, the copula function is selected from 24 candidate families following the instructions provided in Moftakhari et al., 2017, using the R-package "copula" (Kojadinovic & Yan, 2010)."

R2C4:

4.   Line 53: change "not available" to "unavailable"

Author Response:

We have revised it as suggested.

R2C5:

5.  Line 87: "has accounted"

Author Response:

We have revised it as suggested.

R2C6:

6.  Line 108-111: Why did the authors use GRFR runoff forcing? ERA5 land and NLDAS also provide runoff.

Author Response:

Thanks for the comment.

The GRFR runoff forcing is a well-calibrated and bias-corrected global dataset, which has shown excellent performance of simulating extreme streamflow events (Yang et al., 2021). The NLDAS land forcing was also tested but resulted in lower model performance. It remains a major challenge to quantify the uncertainty in runoff and streamflow simulation of Earth system modeling. Fully addressing such uncertainty is beyond the scope of this study. In response to this comment, we elaborated on the selection of the GRFR forcing:

"The GRFR forcing has shown excellent performance of simulating extreme streamflow events (Yang et al., 2021)."

and added a remark for this uncertainty in the discussion (Section 4.2):

"As quantifying the uncertainty of simulated runoff and streamflow in ESMs remains challenging (Lawrence et al., 2019), fully addressing such uncertainty in CFRAs is beyond the scope of this study."

R2C7:

7.  Fig.7's caption is unclear. "Figure 7: The Kendall's correlation coefficient ($\tau$)." What is this $\tau$ and how to define it?

Author Response:

We apologize for the confusion. The caption of Figure 7 (now Figure 10) is edited with more details. Please note that the order of the figures is changed as we split each of the original Figure 3, 4 and 6 into two separate figures to increase clarity.

We have also changed the color scheme to a perceptually uniform one as suggested by the other reviewer. We added the label for the colormap in subfigure (a) and added the x and y labels in the subplot of subfigure (b). The subfigure (d) has been removed in all spatial maps.

[Figure]

**Figure 10:** The Kendall's correlation coefficient ($\tau$) computed for each MOSART coastal cell using the corresponding $Q$ and $SS$ (Section 2.1). The insert in subplot (b) illustrates the basin-averaged $\tau$ provided in the counter-clockwise order of the river basins along the West, Gulf and East coastlines with the error bars representing the corresponding standard deviation.

R2C8:

8.  Fig.8's caption is also unclear. Please specify each term and variable for ($PQ,SS$) in the figure.

Author Response:

Thanks. We added more information in the caption of Figure 8 (now Figure 11) to increase clarity.

[Figure]

**Figure 11:** The joint exceedance probability ($P_{Q,SS}$) computed for each MOSART coastal cell using Eq. 3 (Section 2.1). The insert in subplot (b) is the basin-averaged $P_{Q,SS}$ provided in the counter-clockwise order of the river basins along the West, Gulf and East coastlines with each error bar representing the corresponding standard deviation.

R2C9:

9.      In terms of storm surge events, is it possible to identify storm surge caused by tropical cyclones, extratropical cyclones etc.? How well is GTSM data validated by observations? Given the GTSM products are forced by ERA5 data, it would be useful to validate the GTSM's storm surge.

Author Response:

We appreciate the reviewer comment.

The storm surge events are selected using a peak detection algorithm (Feng et al., 2022), which is based on the signal processing technique and extracts the extreme event that has a peak level over 95th percentile. This method cannot distinguish between tropical cyclones and extratropical cyclones, as the

magnitude of storm surge is influenced by various characteristics of the TC event, such as air temperature, air pressure, wind speed, and wind directions, and their interaction with local environments (Lin and Chavas, 2012).

The total water level has been validated globally in a previous study using gauged data (Muis et al., 2020). Its good performance is also implied in our supplementary information. The storm surge is validated by Dullaart et al., 2020 using several historical events driven by both tropical and extratropical cyclones. This information is added to the revised manuscript to increase clarity.

"The total water level in GTSM has been validated globally against gauged measurements (Muis et al., 2020) and the storm surge has been validated using historical events driven by tropical cyclones and extratropical cyclones (Dullaart et al., 2020)."

R2C10:

10.    Figs 2-4: I would suggest that the authors add labels for geographical locations/states. It is hard to know where the locations are in these figures.

Author Response:

Thanks for the suggestion. We added labels of the state names to Figures 2, 3 and 5 of the revised manuscript.

[Figure]

**Figure 2: Study domain and observations overlaid on the USGS 3D elevation map (U.S. Geological Survey, 2019). The black and blue circles represent the USGS and NOAA gauges, respectively. The gauges used for identifying uncertainties are labeled with the gauge ID. The black solid lines are the coastal river network that consists of at most seven cells from each river outlet.**

[Figure]

**Figure 3: The relative riverbed elevation along the (a) West coast, (b) East coast and (c) Gulf coast. The river networks within the MOSART coastal cells are shown as black solid lines.**

[Figure]

**Figure 5: The flow time of river discharge at the MOSART coastal cells along the (a) West coast, (b) East coast and (c) Gulf coast. The river networks within the MOSART coastal cells are shown as black solid lines.**

R2C11:

11.   While it is not reported in this work, the authors could add discussions on the compound flood risk in the future given the climate projection.

Author Response:

We appreciate the reviewer's insight and have provided a discussion on the impacts of climate change on the CF risks at the end of Section 4.1.

"The broader definition of CF can be expanded to include the interaction between CF drivers and climate drivers (Zscheischler et al., 2020). Global warming will likely increase the frequency of extreme precipitation (Alfieri et al., 2016), the intensity of river discharge (Bermúdez et al., 2021) and storm surge

(Camelo et al., 2020), and the duration of the fluvial and coastal flooding (Feng et al., 2022). All these factors contribute to the exacerbation of CF risks, as both marginal and joint exceedance probabilities will increase. Moreover, climate change has the potential to alter the characteristics or distributions of CF drivers. For instance, the dependence between storm surge and precipitation is enhanced by climate warming, which increases the CF hazard (Wahl et al., 2015). The elevated sea level will move the backwater extent further upstream, increasing the CF exposure (Kulp and Strauss, 2019). Given the uncertainty of climate change, Earth System Models (ESMs) should be increasingly used to understand the potential impacts of different socioeconomic pathways on the CF risk."

R2C12:

12.  In Section 3.2 CFRA, it would be better to add more discussions on the results across different coastal regions and the reason why there is spatial discrepancy in the results such as joint exceedance probability. Is the discrepancy in the risk across different regions due to differences in extremes Q or in SS (e.g., storm type, frequency or track), or in both?

Author Response:

Thanks. We added a new paragraph to discuss the inter-and intra- variabilities of the CF hazard and the reasons. Please note that Figures 4, 6, 8 and 10 are the previous Figures 3, 4, 6 and 8, respectively.

[revised manuscript text omitted]

---

## Author Comment (AC3)

Response to Reviewers

Title: Understanding the Compound Flood Risk along the Coast of the Contiguous United States

Author Response 1st revision

Reviewer 3
Reviewer Comments:

The paper assesses compound flooding for CONUS based on a statistical analysis of compound flood drivers and an analysis of population exposed to flooding from coastal backwater. It also investigates the skill of the model to represent observed peak timing as well as the correlation between, and (joint) exceedance probabilities of surge and discharge. The paper potentially provides an interesting contribution to current literature. However, I have some major comments that I would like to see addressed before the paper is accepted for publication.

Author Response:

We would like to sincerely thank the reviewer for the valuable comments and recommendations. We have carefully addressed the reviewer's suggestions as follows. Excerpts of the revised manuscript are provided to explain our responses to the review comments, as HESS does not allow us to share our revised manuscript at this stage of the review process.

R3C1:

The paper would really benefit from a comparison, both in the introduction and discussion on results, with other large-scale coupled river-coast models (e.g. Ikeuchi et al., 2017 and Eilander et al., 2020), CONUS flood risk models (e.g. Bates et al., 2021) and statistical CF analysis (e.g. Wahl et al., 2015 and Nasr et al., 2021).

Author Response:

We appreciate the valuable comment and references. This work is not only inspired by and but also builds upon these studies that either proposed the statistical approaches or performed large-scale simulations to quantify the CF risk from basin to global scales. Although these studies have already been acknowledged in our literature review, we agree with the reviewer that more thorough discussions should be provided in the introduction to compare the existing studies and in the discussion to better explain the difference of the resulting CF risks when using different approaches.

In the introduction of the revised manuscript, we first elaborated on the literature review of the statistical approaches with a brief comparison of the listed studies:

"At regional and global scales, statistics-based CFRAs consider the CF hazard as statistical dependence or co-occurrence rate of multiple flood drivers including discharge and surge (Moftakhari et al., 2017; Sadegh et al., 2018; Muñoz et al., 2020), precipitation and surge (Bevacqua et al., 2019), discharge, surge, and wave (Camus et al., 2021), etc. Statistics-based CFRAs perform statistical analysis using long-term data at paired gauges near the land-ocean interface. The data can be obtained either from large-scale numerical simulations (Eilander et al., 2020; Nasr et al., 2021) or gauge observations (Ward et al., 2018; Paprotny et

al., 2020). Bivariate or multivariate analyses are performed to measure the CF hazard in terms of the joint occurrence of event extremes (Salvadori et al., 2007; Zscheischler et al., 2020). The CF hazard is determined either using the extreme dependence among multiple CF drivers or the likelihood of their joint occurrence. The dependence structure can be assessed from correlation and/or tail dependence coefficients (Wahl et al., 2015; Nasr et al., 2021). The co-occurrence rate may be calculated as the joint exceedance probability when a single or multiple drivers are above their predefined thresholds (Moftakhari et al., 2017), e.g., 95th or 99th percentile (Kew et al., 2013), which are defined respectively as "OR" and "AND" hazard scenarios by Salvadori et al., 2016."

We elaborated the existing studies of large-scale models and acknowledged a particular contribution from the large-scale coupled river-coast models.

"Such models offer the capability to evaluate spatiotemporally varied CF drivers, flood extent and population exposure to CF events from basin to global scales over multiple decades (Ikeuchi et al., 2017; Eilander et al., 2020; Eilander et al., 2023)."

The existing CFRA studies in the CONUS domain was briefly discussed in the last paragraph of the introduction. We elaborated this part with more details added. in the discussion.

"The contiguous United States (CONUS) (Fig. 1) consists of 48 states, with coastal counties occupying about 10% of the total area. There are 17 major port cities, and ~40% of the US population residing in coastal counties are subject to high coastal flooding risks (Hanson et al., 2011). A high-resolution analysis study including pluvial, fluvial, and coastal flooding, projected a significant changing pattern of the flood risk in CONUS under future climate scenarios (Bates et al., 2021). Particularly, the CF risk was previously evaluated for the CONUS coastline or major US coastal cities using statistics-based CFRAs in terms of the dependence between storm surge and precipitation (Wahl et al., 2015), seasonable dependence among multiple CF drivers (Nasr et al., 2021), and the joint probability in "OR" hazard scenarios in response to sea level rise (Moftakhari et al., 2017)."

Finally in the result discussion (the end of Section 3.2.1), we provided a comparison of the risk assessments between ours and other statistics-based CFRAs. We highlighted that the CF risk could vary significantly when considering different CF drivers or using different statistical approaches.

"The CF hazard computed in this study shows both similarities and notable differences with previous statistics-based CFRAs (Eilander et al., 2020; Nasr et al., 2021; Wahl et al., 2015). For example, our analysis reveals several localized hotspots of the CF hazard characterized by a strong dependence between $Q$ and $SS$ in the Northwest and Gulf coasts (Fig. 12), as indicated by Eilander et al. (2020). However, the calculated $\tau$ values in our study are generally lower than those computed using annual maxima sampled from $Q$ and $SS$ observations in the East coast (Nasr et al., 2021). Also, our $\tau$ values are higher than that derived from the dependence of $Q$ and precipitation along the West coast, which study also demonstrates substantial variations in $\tau$ at specific locations when using different sampling approaches for the two CF drivers (Wahl et al., 2015). These differences result from variations in the sampling of extreme events, the specific CF drivers considered, the statistical methods employed, as well as other uncertainty sources discussed in Section 2.2.1. Despite the variations observed among different frameworks, each study provides unique insights into the understanding and addressing the complexities associated with CF risks. The choice of a specific CFRA depends on the local characteristics of the study area and the specific requirements of local flood planning and management."

R3C2:

The difference between "data-based CFRA" (line 40) and "physics-based CFRA" (line 60) is not clear but essential to understand the introduction. It would help to start with defining both concepts before discussion pros and cons of both approaches. My first guess was that physics-based would refer to numerical models and data-based to observations, but models are also mentioned under data-based CFRAs (line 45) and observations under the physics-based approach (line 61). Also, the approach of Ikeuchi et al. 2017 and Eilander et al. 2020 are basically similar (analysis of simulated estuarine water levels in a coupled CaMa-Flood and GTSM model) but here mentioned as different approaches.

Author Response:

Thanks for the comment.

According to the suggestion from another reviewer, we changed the classification of CFRAs from "data-based" and "physics-based" to "statistics-based" and "dynamics-based" throughout the revised manuscript. As "statistics-based" CFRA better pertains to the approaches that rely on the statistical analysis and data interpretation to understand the CF risk, where data can be either from observations or numerical models. "Dynamics-based" refers to the CFRA that, as the reviewer pointed out, is based on numerical simulations that could represent how systems change over time and how various factors interact to determine the risk patterns.

As suggested by this comment, we have clearly defined both concepts before discussing the details of the two approaches:

"CFRA can be classified into statistics-based and dynamics-based approaches. Statistics-based CFRAs rely on statistical modeling and define the CF hazard as the frequency of a CF event. Dynamics-based CFRAs use numerical simulations that can represent the spatiotemporal variabilities of CF drivers and how various CF drivers interact."

Yes. Ikeuchi et al. 2017 and Eilander et al. 2020 used similar modeling approaches as we introduced in this work. This study is inspired by their great work. We classify Ikeuchi et al. 2017 into dynamics-based CFRAs as their work mainly focuses on the dynamics of inundation simulated using the coupled numerical models. And we consider Eilander et al. 2020 as a key reference in which data used in statistical analysis can be sourced from large-scale numerical simulations, as in their work the dependence among CF drivers is assessed using the modeled variables.

R3C3:

The terms "hazard risk" and "exposure risk" (line 73) are confusing in my opinion as risk is usually referred to as the combination of hazard, exposure and vulnerability. "hazard risk" seems to only consider the hazard and seems similar to what Couasnon et al. (2020) call "compound flood potential". In short, I think usage of the term "risk" here is confusing and the term "comprehensive risk" (line 98) is even a bit misleading.

Author Response:

We appreciate the reviewer comment. Our intention of categorizing the CF risk into the hazard risk and exposure risk was because previous studies only focused on one particular type of the risk and we were trying to distinguish the two categories and propose the idea of integrating them into a single measure. Although flood exposure risk and flood hazard risk are defined elsewhere in the literature, we agree with the reviewer that such terminology is used less frequently than simply "flood exposure" and "flood hazard". Similar to flood potential, flood hazard is commonly used to define the risk associated with the frequency of a flood event.

As the reviewer mentioned, flood risk usually refers to the combination of hazard, exposure and vulnerability. However, it is sometimes challenging to quantify vulnerability as it subjectively depends on a few factors (e.g. exposure, sensitivity, adaptive capacity and resilience) and the availability of such data is usually limited in large-scale domains. Thus, we consider the CF risk as a combination of hazard and exposure, similar to the risk defined as flood exceedance probability and damage in Judi et al., 2018 and Kalyanapu et al., 2015.

In response to this comment, we replaced "CF hazard risk" and "CF exposure risk" with "CF hazard" and "CF exposure" throughout the revised context. "CF comprehensive risk" is changed to "CF risk".

R3C4:

The computation of the "risk" metrics is not entirely clear due several ambiguities in the methods section. Specifically for the sampling of events for the calculation of both metrics, see specific comments below.

Author Response:

We apologize for the confusion. Please see the responses to the following comments where we provided a clearer description of the methodology.

R3C5:

The Figures with maps and the overlying bars or dots are very difficult to read (Fig 3, 4 & 6).In Fig 6, 7 and 8 the color bars don't have a title or unit and there is no legend for the colors or x-axis label for the subplots.

Author Response:

We apologize for the unclear figures. The original intention of overlaying bars/dots on contour maps is to help readers visualize relevant information in a single figure. But we agree with the reviewer that condensing too much information in one figure will only make it harder to read.

In the revised manuscript, each of Figure 3, 4 and 6 is split into two separate figures. We also removed the subplot d (i.e., the overview of the CONUS domain) as this information is already presented in Figure 2. The original Figure 3 is divided into Figure 3 and 4. Figure 3 shows the spatial map of the riverbed elevation and Figure 4 shows the histogram of relative importance of the three CF drivers. The numbers representing river basins are marked in Figure S3 of the supplement. The original Figure 4 is divided into Figure 5 and 6. Figure 5 shows the spatially-varied flow time and Figure 6 shows the bars that represent

the shifted days in $Q$ and $SS$ peaks. The numbers representing the observation gauges are marked in Figure S4 of the supplement. The original Figure 6 is divided into Figure 8 and 9. Figure 8 is the marginal exceedance probability $P_Q$ computed for all MOSART coastal cells and Figure 9 is the marginal exceedance probability $P_{SS}$ computed from the GTSM cells nearest to the corresponding outlets. The color scheme of the spatial maps is changed to a perceptually uniform one. State names are added to Figure 2, 3 and 5.

In Figures 8~11, we added labels to the color bars and x- and y- axis labels to the subplots, as well as more detailed description in the caption.

[Figure]

Figure 3: The relative riverbed elevation along the (a) West coast, (b) East coast and (c) Gulf coast. The river networks within the MOSART coastal cells are shown as black solid lines.

[Figure]

**Figure 4: The relative importance of *Q* (red), *SS* (blue) and relative riverbed elevation (gray) provided in the counter-clockwise order of the river basins along the West, Gulf and East coastlines. The numbers representing individual river basins correspond to those in Figure S3.**

[Figure]

**Figure S3.** The numbers representing individual river basins corresponding to those in Figure 4.

[Figure]

**Figure 5: The flow time at the MOSART coastal cells along the (a) West coast, (b) East coast, and (c) Gulf coast.**

[Figure]

**Figure 6: The shifted days in the $Q$ and $SS$ peaks between the observation gauges and the corresponding river outlets provided in the counter-clockwise order of gauges along the West, Gulf and East coastlines. The rectangular box represents the averaged shift over the simulation period and the error bar represents the standard deviation. The numbers representing observation gauges correspond to those in Figure S4.**

[Figure]

**Figure S4.** The numbers representing USGS gauges (red) and NOAA gauges (blue) corresponding to those in Figure 6.

[Figure]

Figure 8: The marginal exceedance probability $P_Q$ along the (a) West coast, (b) East coast and (c) Gulf coast. The basin-averaged $P_Q$ is provided in the counter-clockwise order of the basins along the US coast in the lower left insert of subplot (b) where the error bars represent the corresponding standard deviation.

[Figure]

Figure 9: The marginal exceedance probability $P_{SS}$ located at the GTSM cells nearest to the corresponding outlets.

[Figure]

**Figure 10:** The Kendall's correlation coefficient ($\tau$) computed for each MOSART coastal cell using the corresponding $Q$ and $SS$ (Section 2.1). The insert in subplot (b) is the basin-averaged value of $\tau$ provided in the counter-clockwise order of the river basins along the West, Gulf and East coastlines with the error bars representing the corresponding standard deviation.

[Figure]

**Figure 11:** The joint exceedance probability ($P_{Q,SS}$) computed for each MOSART coastal cell using Eq. 3 (Section 2.1). The insert in subplot (b) is the basin-averaged value of $P_{Q,SS}$ provided in the counter-clockwise order of the river basins along the West, Gulf and East coastlines with each error bar representing the corresponding standard deviation.

Specific comments

R3C6:

line 28: "A CF event [...] occurs when the associated drivers exceed their respective thresholds (Zscheischler et al., 2020)." Zscheischler et al. (2020) actually argue that not all associated drivers need to exceed their respective thresholds to have large impact CF events. I suggest to rephrase this sentence.

Author Response:

We apologize for the confusion. We tried to express the same concept as explained by the reviewer. This sentence is now rephrased as: "It is possible that a compound flood (CF) event is not caused by extreme

weather (Couasnon et al., 2020) but rather occurs when one or multiple flood drivers exceed their respective thresholds (Zscheischler et al., 2020)."

In our analysis, a CF event is identified when both drivers (i.e., $Q$ and $SS$ in our case) exceed their thresholds because it is otherwise difficult to identify the CF events in a large-scale analysis. We appreciate the reviewer's remark. To further acknowledge this limitation, more discussions have been added to Section 4.2.

"Second, we quantify the CF impacts using the SS-induced backwater effects without considering the complex interactions between the two flood drivers or the possibility of a CF event induced by an individual extreme driver. CF does not necessarily require all drivers to exceed their corresponding thresholds (Zscheischler et al., 2020)."

R3C7:

line 78: Do you mean "A robust CFRA should provide a thorough understanding of the uncertainties related to the risk analysis such as uncertainties associated with flood frequency and possible flood damages" ? Would it be possible to make this more specific for *Compound Flood* Risk Analyses?

Author Response:

Thanks for the insightful suggestion. This sentence is rewritten to highlight uncertainty analysis for CFRA.

"A robust CFRA should consider the uncertainties associated with frequency and possible damages of compound flooding and provide a thorough understanding of the uncertainties related to the risk analysis (Apel et al., 2004). The uncertainty can stem from various sources in both statistics-based and dynamics-based CFRAs, such as measurement errors and approximations in numerical models. A comprehensive understanding of the uncertainty sources in CFRA is crucial for managing and predicting CF risks and will provide valuable insights for guiding future improvements."

R3C8:

line 89: Please clarify what is meant by "variability in the fluvial process".

Author Response:

By "variability in the fluvial process", we refer to the variations in the characteristics or behaviors of rivers over time and space. More specifically in this study, the variability is represented by the spatiotemporally varied streamflow.

Thanks for the suggestion. We elaborated this sentence to increase clarity.

"However, most existing studies rely on the flood driver measured/modeled at a single site and have not accounted for the dynamic change of river flow, such as the spatiotemporally varied streamflow, as well as river topology, coastal backwater effects and the associated uncertainties."

R3C9:

line 103-118 (CFRA model framework): could you provide some more details of the models. I.e. what is the temporal resolution of the MOSART and GTMS model outputs used here?  How are the channel widths, depth and lengths defined? What equations are solved in the model (Full Saint-Venant, Local inertial, other)?

Author Response:

We appreciate the reviewer comment.

The statistical analysis uses daily streamflow from MOSART and daily maximum $SS$ from GTSM. This is specified as: "The simulated daily streamflow ($Q$) at each selected cell is paired with the daily maximum storm surge ($SS$) level from the GTSM reanalysis dataset at the grid cell nearest to the outlet."

The temporal resolutions of MOSART and GTSM model outputs are daily and hourly, respectively. This information has been added to the revision:

"The model is run at an hourly time step from 1979 to 2018 and daily outputs are archived for analysis."

"Driven by the ERA5 atmospheric reanalysis dataset, the GTSM produces time series of hourly total water level and storm surge at global coasts from 1979 to 2018 (Muis et al., 2020),"

We agree with the reviewer that MOSART configuration and channel parameters are critical for the simulation. This study uses similar configuration in the same domain from a few previous studies and this information was specified as

"For more detailed descriptions of the model, please refer to Li et al., 2022."

We only described the specific changes in model configuration that are different from this reference, including the implementation of a macro-scale inundation scheme (Luo et al., 2017) and the updated river channel slope derived from a higher resolution DEM, and the downstream boundary conditions.  We did not provide more details because we considered the focus of this study as risk assessment and uncertainty analysis.

In response to this comment, we added more details in model configuration and provide more references:

"MOSART is a physics-based river routing model at the basin to global scales. The model takes the total runoff generated by a land surface model and routes the surface runoff from hillslope to tributary subnetworks, which along with the subsurface runoff are discharged to river outlets through the main channels. In this study, kinematic wave method is used for overland flow routing, whereas diffusive wave method is applied in the river channels to represent the coastal backwater effects (Feng et al., 2022). The MOSART simulation is performed on the 1/8° resolution CONUS grid. The MOSART configuration on the same grid has been validated and applied to flow and sediment simulations (Li et al., 2015a; Li et al., 2022). The model parameters are available globally with more detailed descriptions in previous studies (Li et al., 2013; Li et al., 2015b). The model is run at an hourly time step from 1979 to 2018 and daily outputs are archived for analysis. The first-year simulation is excluded for analysis due to model spin-up. The floodplain inundation is represented using a macroscale inundation scheme (Luo et al., 2017)."

R3C10:

line 107: Is the slope of a 15 arcsec DEM (~25km) adequate to estimate the channel river slope?

Author Response:

We appreciate the reviewer's comment. We note the resolution of 15 arc-seconds is approximately 500 meters at the equator, which is adequate for resolving topology in the 1/8° MOSART grid.

R3C11:

line 133: Towner et al. (2019) study the skill of several GFMs (not including MOSART or GTSM) for peak flows in the Amazon, how does that relate to the skill of your model framework?

Author Response:

We apologize for any confusion.

We are aware that Towner's study is applied to a different domain using different GHMs. This study has performed a comprehensive skill assessment of various GHMs, providing reasonable guidance for evaluating the GHM performance in large-scale simulations. We used this reference to support that the MOSART performance is reasonable in the large-scale simulation as the skill metrics meet the standards proposed in Towner et al. 2019.

In the revision, this sentence is rewritten and the reference is deleted to avoid confusion:

"In the context of constructing a new CFRA framework within the CONUS domain and investigating the associated uncertainties, the performance of MOSART and GTSM models is deemed satisfactory in large-scale simulations."

R3C12:

line 139 (step a): The sampled events should be independent. How is this ensured?

Author Response:

We apologize for the typo and the missing information regarding this comment.

The first step in calculating the CF hazard should be referred as "CF event selection" instead of "storm surge event selection". In the sampling CF events, we first extract $SS$ events using the event selection scheme proposed in Feng et al., 2022. This scheme uses a peak detection algorithm to filter independent $SS$ events. Within each selected $SS$ event, a CF event is identified if $Q$ is over the predefined threshold of 95th percentile. The advantage of this method is that it allows a more realistic representation of a $SS$ event than defining a time window around the peak, as $SS$ may last longer than a few days.

This method also ensures that the selected $SS$ events are independent as the sign of water level is always changed even between two adjacent events that are close to each other. While this method does not

ensure the independence of $Q$, the frequency of a $SS$ event is generally smaller than that of fluvial flooding.

In response to this comment and R3C13, we elaborated on the description of step (a) in the CF hazard calculation and explanations of independence to increase clarity.

"(a) CF event selection: use a $SS$ event selection scheme (Feng et al., 2022) to extract all $SS$ events with the $SS$ level over 95th percentile and then in the selected SS events identify them as CF events if river discharge of the corresponding station during these events is also over 95th percentile;"

"As the first step, our event selection scheme samples independent $SS$ events from the time series data, which avoids dependence in the extremes and does not require declustering. We assume that the $Q$ extreme within each $SS$ event is independent as both frequency and duration of $SS$ are generally smaller than that of fluvial flooding."

R3C13:

line 142 (step b): How are Q events sampled? These are not mentioned under step a.

Author Response:

Please see the response to R3C12.

R3C14:

line 144 (step c): It is not clear how bivariate variables are defined. Is this based on AND or OR sampling of the variables? And do you allow for any time lag between the variables? Please clarify.

Author Response:

We appreciate this comment.

The bivariate variables are defined based on the "AND" hazard scenario (Salvadori et al., 2016). This may be implied by Equation 3. We have it clearly defined in the revised manuscript:

"(d) bivariate analysis: calculate the joint exceedance probability based on "AND" hazard scenario (Salvadori et al., 2016) that accounts for both marginal distributions and dependence structure."

We acknowledge that the duration of a fluvial flood event may not precisely align with the duration of $SS$ during a compound flood (CF) event. It is possible there is small overlapping between a $SS$ event and the corresponding fluvial flooding event. But we don't allow any lag for Q because the CF exposure is measured as the extent where $SS$-induced backwater matters. Such effects are dominated by $SS$ and will only be significant during the selected $SS$ events. This discussion is added to the revision.

"While it is possible that the duration of a fluvial flood event does not precisely align with a $SS$ event, we don't include any time lag between $Q$ and $SS$ in the consideration as this study specifically quantifies the CF impact based on the $SS$-driven backwater effects."

R3C15:

line 179-184 (exposure risk): The exposure risk metric accounts not only for surge but also tide as the 'baseline' downstream boundary conditions is based on MSL only (and not MSL+tide). This has several consequences on the analysis in my opinion which are not included nor discussed. For instance, the tidal amplitude is in many locations probably an important predictor of backwater volumes (section 2.2.2) but not accounted for. It could also explain some of the differences between both CF metrics which is not discussed (section 3.2.2). And it should not be referred to as "surge-induced backwater effects" (line 405).

Author Response:

Thanks for the comment. Tide is excluded from our simulations. As defined in Section 2.1, at the downstream boundary of MOSART only the time series of storm surge are applied.

"We apply two types of boundary condition (BC): (1) time-varying storm surge ($SS$) level and (2) fixed mean sea level. Both are obtained from the third-generation Global Tide and Surge Model (GTSM) (Muis et al., 2022). The $SS$-induced backwater effects in this study are quantified by comparing the two simulations which use the first and second BCs, respectively (Feng et al., 2022)."

We have not accounted for tide because: (a) the higher-frequency variability of tides compared to river discharge and storm surge poses challenges in quantifying tide as a CF driver along with the other two; (b) our sampling algorithm is only able to extract the low-frequency $SS$ but not tides; and (c) we previously showed storm surge dominates the backwater effects in a low-lying river basins (Feng et al., 2022). Thus, this study only considers the $SS$-induced backwater effects. But we agree with the reviewer that tide and its nonlinear interaction with storm surge could be an important predictor in many locations. This limitation is now acknowledged in Section 4.2.

"The backwater effects are driven in MOSART by prescribing the $SS$ time series at the downstream boundary. However, the actual CF is driven by the interactive processes between multiple drivers, including precipitation, land surface runoff and inundation, river discharge and coastal tide, storm surge and wave (Nasr et al., 2021), as well as their nonlinear interactions. For example, the interaction between flooded water and channel flow, groundwater and surface water, river discharge and upstream propagation of ocean tides and storm surge will likely attenuate the hydrograph, intensify inland flooding or affect the backwater propagation, particularly in low-lying watersheds. Such interactions contribute to another layer of complexity and uncertainty at the terrestrial and aquatic interface and should be simulated using ESMs with fully coupled land, river and ocean modeling components."

R3C16:

line 193: "the CF hazard index (CFHI) and the CF exposure index (CFEI) are obtained by normalizing Pq,ss and Wp with their corresponding 95th percentile values". How are the 95th percentile values calculated? If I understand correctly, both indicators are a single value per cell right?

Author Response:

Yes. The 95th percentile value is calculated from every grid cell. This is clarified in the revision:

"the CF hazard index ($CFHI$) and the CF exposure index ($CFEI$) are obtained by normalizing $P_{Q,SS}$ and $W_p$ with their corresponding 95th percentile values at every grid cell."

R3C17:

line 196: "Our approach transforms the probability of occurrence into a direct measure of human exposure." Could you explain how?

Author Response:

This sentence is rephrased to:

"Our approach integrates measures of risks that consider both the probability of occurrence and human exposure."

R3C18:

line 239-250 (Impact of riverbed elevation): Is this analysis done per cell or per basin? And where does the riverbed elevation data come from? (see also earlier comment on the CFRA framework)

Author Response:

This analysis is performed for every cell. This is explained as "For each coastal grid cell, we use the MOSART simulated $Q$, the GTSM simulated $SS$ at the river outlet, and the grid cell elevation."

The riverbed elevation is derived from the 15 arcsec digital elevation model (DEM) of the HydroSHEDS, which has been clarified:

"the channel slope and the riverbed elevation are derived from the 15 arcsec digital elevation model (DEM) of the HydroSHEDS and river vector data."

R3C19:

line 384: "In summary, CFRA should not rely on any single method; more comprehensive thinking is needed considering the different characteristics among the different risk types." How does the CF comprehensive risk metric compare to an actual risk analysis (i.e., combining the hazard and its potential consequences to derive e.g., annual expected losses or people exposed)

Author Response:

We appreciate the reviewer's insight. This study does not intend to critique any existing approaches. Rather, we build on their success and aim to demonstrate that more factors can be integrated to provide a comprehensive understanding of the CF risk.

This sentence is rephrased to: "to comprehensively understand the complex CF risk, it is critical for CFRAs to integrate multiple risk factors based on the available approaches."

Based on our experience, the actual flood risk analysis is a real-time or near real-time assessment in specific areas that updates risk factors based on the latest available data. Such analysis typically has a different focus from ours. While the goal of the actual risk analysis is to provide timely and accurate information to emergency responders, the framework proposed in this study aims to assess the risk at the continental scale to understand the spatial variability and uncertainty within different risk factors, and can help, e.g., "provide target regions where the computational mesh should be refined to improve model accuracy." Instead of delving into excessive details of comparing metrics, we acknowledge in Section 2.1 that "the combination of different types of risks, despite providing a comprehensive estimation of the CF risk, is subjective and may affect the risk assessment results."

Bates, P. D., Quinn, N., Sampson, C., Smith, A., Wing, O., Sosa, J., Savage, J., Olcese, G., Neal, J., Schumann, G., Giustarini, L., Coxon, G., Porter, J. R., Amodeo, M. F., Chu, Z., Lewis-Gruss, S., Freeman, N. B., Houser, T., Delgado, M., Hamidi, A., Bolliger, I., McCusker, K., Emanuel, K., Ferreira, C. M., Khalid, A., Haigh, I. D., Couasnon, A., Kopp, R., Hsiang, S., and Krajewski, W. F.: Combined modeling of US fluvial, pluvial, and coastal flood hazard under current and future climates, Water Resour. Res., 57, e2020WR028673, https://doi.org/10.1029/2020wr028673, 2021.

Couasnon, A., Eilander, D., Muis, S., Veldkamp, T. I. E., Haigh, I. D., Wahl, T., Winsemius, H. C., and Ward, P. J.: Measuring compound flood potential from river discharge and storm surge extremes at the global scale, Nat. Hazards Earth Syst. Sci., 20, 489-504, https://doi.org/10.5194/nhess-20-489-2020, 2020.

[revised manuscript text omitted]

---

## Author Response (AR2)

Response to Reviewers

Title: Understanding the Compound Flood Risk along the Coast of the Contiguous United States

Author Response 2nd revision

Reviewer 1

Reviewer Comments:

R1C1:

L39-42: I think the definition of the now-called statistics and dynamics-based approaches is still not clear enough from the outset. Specifically, the authors write that statistical-based CFRA defined the CF hazard as the frequency of a CF event. Could the authors clarify the difference with how dynamics-based CFRA would define CF hazard? Additionally, dynamics-based CFRA is defined to represent spatiotemporal variabilities of CF drivers -> what 'spatiotemporal variabilities' are meant that cannot also be captured with a statistical analysis is also unclear. As these definitions play an important role in the paper, I think it is important that they are made more clear. Perhaps the denominators 'statistical' and 'dynamic' are not 100% satisfactory after all as the authors also contrast the methods in terms of them considering only the CF hazard v.s. the CF risk.

Author Response:

We appreciate the reviewer comment. In the revision, we rephrased "dynamics-based" to "hydrodynamics-based", which is a more accurate (despite less generalized) term representing the method based on numerical model simulations and fits better for this manuscript.

The hydrodynamics-based CFRA is used to assess the CF exposure. This is mentioned later as (L78-79): "The CF hazard and exposure evaluated separately by the aforementioned statistics-based or hydrodynamics-based CFRAs may produce inconsistent results (K. Xu et al., 2022)." We apologize for the confusion and have rewritten this sentence to improve clarity.

"Hydrodynamics-based CFRAs use numerical simulations that can represent human exposure to CF events with considering the spatiotemporal variabilities and interaction of CF drivers."

R1C2:

L395-396: "The choice of a specific CFRA depends on the local characteristics of the study area and the specific requirements of local flood planning and management." Could the authors please give some examples to substantiate this remark?

Author Response:

Thanks for the comment. The discussion of flood planning and management is a separate topic and is not the focus of this manuscript. To avoid confusions, we decided to remove this sentence.

R1C3:

L438: "The broader definition of CF can be expanded to include the interaction between CF drivers and climate drivers" I am unclear why the definition of compound flooding needs to be expanded to study changes in CF due to climate change. Could the authors please clarify this in the text?

Author Response:

We apologize for the confusion. We were trying to explain that CF is affected by climate change with its drivers interacting with climate drivers. This sentence is rephrased to improve clarity.

"CF hazard and exposure can also be impacted by climate change as CF drivers interact with climate drivers."

R1C4:

L439-441: "Global warming will likely increase the frequency of extreme precipitation (Alfieri et al., 2016), the intensity of river discharge (Bermúdez et al., 2021) and storm surge (Camelo et al., 2020), and the duration of the fluvial and coastal flooding (Feng et al., 2022)" - please specify in which regions, as this is not true everywhere?

Author Response:

We agree that this may cause confusions and have added "in many regions" and an example of US east coast to generalize the impact of global warming:

"Global warming will likely increase the frequency of extreme precipitation (Alfieri et al., 2016), the intensity of river discharge (Bermúdez et al., 2021) and storm surge (Camelo et al., 2020), and the duration of the fluvial and coastal flooding (Feng et al., 2022) in many regions, such as the US east coast (Ting et al., 2019)."

R1C5:

L445-447: why necessarily using ESMs? Or do you mean that more research on the impact of climate change on CF is needed?

Author Response:

Yes. We rephrased this sentence to

"Given the uncertainty of climate change, more attention should be paid to understand the potential impacts of different socioeconomic pathways on the CF risk."

R1C6:

L454-455: "The reanalysis forcing typically covers a shorter period (e.g. from 1979 to 2018) and thus may underestimate extreme events, such as TCs." -> I think this needs some rephrasing to say that statistical

analyses based on such periods may lead to unreliable estimates of the return frequency/probability of TCs

Author Response:

Thanks for the suggestion. This sentence is rephrased as suggested.

R1C7:

L457-L64: this discussion is about the quality of ESMs which isn't obviously connected to the limitation of your study in terms of period length. Do you mean that you would use ESMs to analyze historical CF risk as these offer simulations with longer periods than reanalysis data? Please clarify.

Author Response:

Thanks for your comment. This is exactly one advantage of running ESMs that offers longer-period simulations. This is clarified in the revised manuscript:

"ESMs, which can simulate CF drivers for longer periods than reanalysis data, can be used to analyze historical CF risks."

The other advantage is that ESMs with enhanced capabilities at resolving the extreme events at the coastal zone could potentially provide improved simulations than the available reanalysis dataset. This was mentioned as "Moreover, high-resolution cloud-resolved ESMs show promising performance in representing extreme events (Caldwell et al., 2021), which can help the hydrodynamics-based CFRA quantification."

Textual comments:

R1C8:

L7: 'statistics-based statistical analyses': double statistic, suggest to reword

Author Response:

Agree. This is rephrased to "statistics-based analyses".

R1C9:

L116: 'at the basin to global scales': replace with "which can be applied at basin to global scales"?

Author Response:

This is corrected as suggested.

R1C10:

L123: 'and archives daily output are archived for analysis': needs to be reworded

Author Response:

Sorry for the typo. This is corrected to "and daily outputs are archived for analysis".

Caldwell, P. M., Terai, C. R., Hillman, B., Keen, N. D., Bogenschutz, P., Lin, W., Beydoun, H., Taylor, M., Bertagna, L., and Bradley, A.: Convection‑permitting simulations with the E3SM global atmosphere model, Journal of Advances in Modeling Earth Systems, 13, e2021MS002544, 10.1029/2021MS002544, 2021.

Ting, M., Kossin, J. P., Camargo, S. J., and Li, C.: Past and future hurricane intensity change along the US East Coast, Scientific reports, 9, 7795, s41598-019-44252-w, 2019.